# NOW YOU SEE ME! ATTRIBUTION DISTRIBUTIONS REVEAL WHAT IS TRULY IMPORTANT FOR A PREDICTION

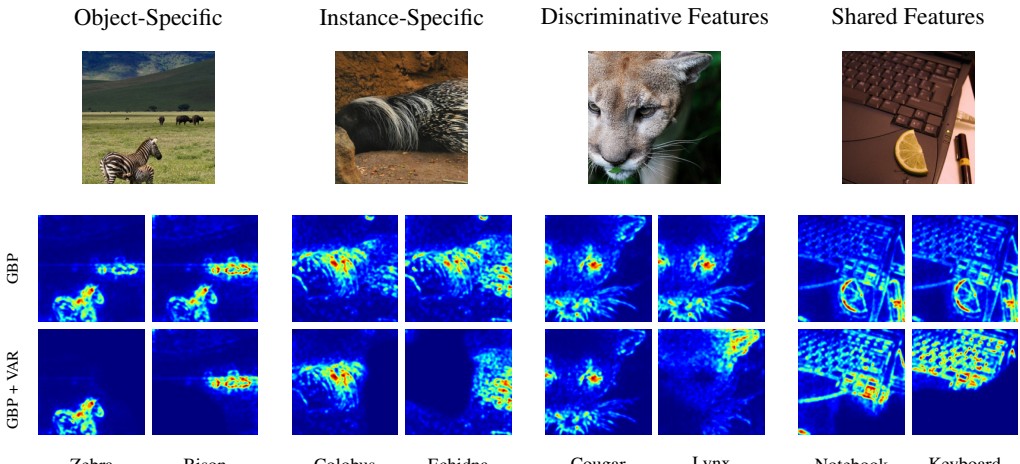

Figure 1: *Attributions on ImageNet.* Attributions computed as distributions across classes are **object-specific** and visually ground correct target objects, are **instance-specific**, identifying features that are relevant on a by-part-basis, and are **class-discriminative**, yielding features that separate closely related classes. They further reveal **shared concepts** between closely related classes. In contrast, the standard approach of computing attributions on the logit of the predicted class does not reveal any of these properties.

## ABSTRACT

Neural networks are regularly employed in high-stakes decision-making, where understanding and transparency is key. Attribution methods have been developed to gain understanding into which input features neural networks use for a specific prediction. Although widely used in computer vision, these methods often result in unspecific saliency maps that fail to identify the relevant information that led to a decision, supported by different benchmarks results. Here, we revisit the common attribution pipeline and identify one cause for the lack of specificity in attributions as the computation of attribution of isolated logits. Instead, we suggest to combine attributions of multiple class logits in analogy to how the softmax combines the information across logits. By computing probability distributions of attributions over classes for each spatial location in the image, we unleash the true capabilities of existing attribution methods, revealing better object- and instance-specificity and uncovering discriminative as well as shared features between classes. On common benchmarks, including the grid-pointing game and randomization-based sanity checks, we show that this reconsideration of *how and where* we compute attributions across the network improves established attribution methods while staying agnostic to model architectures.

## 1 INTRODUCTION

Neural Networks are prime models for decision-making, yet are inherently opaque. Especially in high-stakes prediction, but also in a more general context, there is a growing need for transparent

reasoning that provides the user with an understanding of the model's decision. In the context of Explainable Artificial Intelligence (XAI), explanations that describe *which input features*, such as image regions, are used for a prediction. This group of approaches is coined *attribution methods* as they attribute importance of input features to the output of a model and are among the most popular in XAI and used in high-stakes domains such as medical imaging Borys et al. (2023).

However, it has been shown that these post-hoc explanations have severe shortcomings; while the features often seem sensible, they turned out to not properly model the class-relevant features used by the network Rao et al. (2022a). Thus, the explanations failed in providing the desired information on which input features were actually *relevant* for the classification. Here, we investigate the causes of this shortcoming and find that the **existing attribution methods are actually capable of discovering class-relevant features**, if only we use them right: the problem is how these methods are applied. In models for image classification, attributions are typically computed on the logit of the predicted class. This is, however, not the true classification – the final prediction is determined through the *softmax considering all class logits*. Computing attributions instead directly on the softmax gradient comes with huge numerical issues as the gradient vanishes as networks become more confident. Considering one of the example images in Figure 1, we see that seemingly class unrelated features are highlighted when visualizing a logit in isolation, for example the full animal face for the class lynx. The subsequent softmax of the classification head would, however, consider the cougar logit in the denominator, thus each of its features, here the face, would have a *negative* impact on the classification of the lynx.

To overcome this issue, we suggest to compute *distributions of attributions over multiple classes* in each spatial location (see Fig. 2 bottom). This change in how attribution methods are applied has only little computational overhead, is training-free, applicable to any existing attribution method, and agnostic to architectures. The resulting saliency maps on standard vision benchmarks qualitatively provide much more focused and class-relevant information across different models, including convolutional and Transformer-based architectures (see Fig. 4). Picking up on the lynx example, we see that when we consider the distribution of attributions across classes, the facial features have no relevance for the lynx class, as most of the probability mass is located at the cougar class.

Quantitatively, this reconsideration of the standard attribution pipeline improves the ability to retrieve correctly localized attributions in the grid pointing game Rao et al. (2022a), improves in insertion ablations Kapishnikov et al. (2019), and increases robustness to randomization-based sanity check Adebayo et al. (2018). Hence, our approach enables identifying the actual class-relevant features that a network uses for prediction, supercharging any existing attribution method.

## 2    RELATED WORK

Research in XAI gave rise to three main approaches to discover prediction-relevant input features. Perturbation techniques such as RISE Petsiuk et al. (2018), extremal perturbations Fong et al. (2019), and SHAP Lundberg & Lee (2017) probe model behavior by modifying inputs. While effective, these methods are computationally expensive, often requiring multiple forward passes and significant processing time and often consider input features independently. Approximation techniques, such as LIME Ribeiro et al. (2016) and FLINT Parekh et al. (2021), create interpretable surrogate models to mimic complex networks locally. Such surrogates can, however, largely differ from the target model reasoning, limiting their ability to accurately capture the prediction dependencies.

The third category, activation- and gradient-based attribution methods, strikes a balance between efficiency and fidelity by leveraging the network's internal computation graph. Several approaches have been proposed based on gradients (Input×Gradient, Integrated Gradients, GBP) Simonyan et al. (2014); Sundararajan et al. (2017); Zhuo & Ge (2024); Springenberg et al. (2015) or upsampling of feature maps taking class information into account, such as GradCAM, Selvaraju et al. (2017) or LayerCAM Jiang et al. (2021). A seminal approach considering flow of activation values across the network under a conservation property is Layer-wise Relevance Propagation (LRP) Bach et al. (2015), which however requires architecture-specific adaptations Otsuki et al. (2024); Chefer et al. (2021a). Similarly, DeepLift Shrikumar et al. (2017) uses reference activations to determine neuron importance through custom backpropagation procedures. In the context of transformers, recent approaches focus on the idea of attention roll-out, which reflects the propagation of informa-

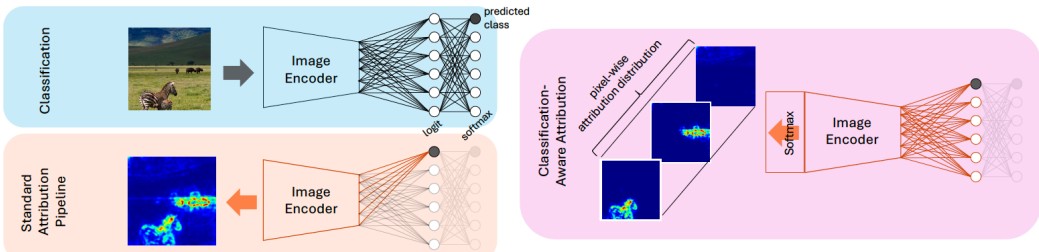

Figure 2: *Reconsidering how to apply attributions.* For an image classification, typically done with a softmax classification head on top of an image encoder (top left), the standard approach to generate attribution maps as explanation for the decision-making is considering the logit of the predicted class (bottom left), ignoring that softmax incorporates logits of all classes for the final prediction. We suggest to compute **distributions of attributions across classes** by computing the softmax of attribution values across **all** logits, reflecting the network decision-making (right). Network parts considered for the attribution computation are colored in orange.

tion through the layers by multiplying each of their transition matrices, including Bi-attn Chen et al. (2023), T-attn Yuan et al. (2021b), and InFlow Walker et al. (2025b).

Because of their widespread use, benchmarking attribution methods in computer vision has been of growing interest. Ancona et al. (2018) study attribution sensitivity and formally proved equivalence between approaches under specific assumptions, wheras Rao et al. (2022b) systematically studied how faithful attributions are to an underlying prediction using the grid-pointing game. Insertion ablations Kapishnikov et al. (2019) instead study the effect of insertion and deletion of attributed pixels on downstream performance as a proxy for attribution quality. Adebayo et al. (2018) evaluate attribution faithfulness based on stability of explanations with randomization of network components, which was later critically revisited Binder et al. (2023). We will use each of these metrics to study the impact of our suggested attribution approach. Orthogonally, different learning objectives have been suggested to generally improve post-hoc explanation quality such as attributions Gairola et al. (2025), which we later relate to our findings.

## 3 RECONSIDERING THE ATTRIBUTION PIPELINE

Post-hoc attribution methods have been shown to perform poorly in recovering the classification-relevant information from the network Rao et al. (2022a); Böhle et al. (2022) and arguably fail network perturbation based sanity checks Adebayo et al. (2018). Commonly, the attributions for a target class–usually the predicted class–are computed using its logit as a target, which, however, means that the attribution will ignore the information from the other logits (see Fig. 2). The actual classification uses all logits, with softmax contrasting the logits between classes, which means we can simply not expect attributions to recover class-relevant features.[1] We, hence, suggest to reconsider this common paradigm and propose to compute attributions for logits of multiple classes and then compute *distributions over those* at each spatial location, similar to how a classification head computes an output distribution over multiple logits. By doing so, we uncover that common attribution methods could retrieve class-specific information; however, they are hidden when looking at individual logits. Before describing this approach more formally, we introduce the necessary notation.

### 3.1 NOTATION

We consider an input $x \in \mathcal{I}$, where here $\mathcal{I} = \mathbb{R}^{H \times W \times d}$ is typically an image of height $H$, width $W$, and $d$ channels. We describe a classification model as a function $S : \mathcal{I} \to \mathbb{R}^C$, where $C$ is the number of classes. The final discrete classification is usually performed as an argmax over $S(x)$. An attribution method provides a map $\mathcal{H} : \mathcal{I} \times S \times \{1, ..., C\} \to \mathcal{I}'$ that for an input, a model,

---

[1] Attribution methods for outputs are usually applied to logits, as numerical issues caused by the flatness of the softmax function at the (important) regions hinder using it directly as a target.

and optionally a target class provides an explanation of a similar shape as the input (for images, we aggregate attributions across the channel dimension). It attributes scores to each feature $i$ in the input describing in how far $S$ uses $x_i$ for the classification (or target class). This broad definition covers all attribution methods discussed in the related work section, examples are Input×Gradient (IxG) as $\mathcal{H}_{\text{IxG}}(x, S, c) = x \odot \frac{\partial S_c}{\partial x}$, or GradCAM as $\mathcal{H}_{\text{GradCAM}}(x, S, c) = \text{ReLU}(\sum_k \alpha_c^k A^k)$, where $\alpha_c^k = \frac{1}{Z} \sum_i \sum_j \frac{\partial S_c}{\partial A_{ij}^k}$ are the importance weights computed by global average pooling of the gradients. We will focus on the typical Computer Vision application in the following, so $\mathcal{I} = \mathbb{R}^{H \times W \times d}$, but note that the underlying idea generalizes to different input domains $\mathcal{I}$.

## 3.2 Deriving a Contrastive Attribution

Standard attribution targets come with a natural drawback. Using a single logit $S_c$ is non-contrastive by construction, since it does not take competing classes into account. A more principled alternative is to consider the softmax probability $p_c$, which inherently contrasts between class logits. However, the gradient of $p_c$, which would provide the attribution signal for most attribution methods, exhibits an important drawback. Let $z_k$ denote the logit $S_k(x)$. The gradient of the softmax probability $p_c$ is

$$\nabla_x p_c = p_c \left( \nabla_x z_c - \sum_{k=1}^{C} p_k \nabla_x z_k \right). \tag{1}$$

While the subtractive term $\sum p_k \nabla_x z_k$ appears to provide the necessary contrast, it is negligibly small in practice. In the high-confidence regime where $p_c \to 1$, the weighted sum of gradients converges to the gradient of the class with the highest logit, $\nabla_x z_c$. This causes the expression in parentheses to approach zero, effectively annihilating the gradient signal. The softmax gradient, therefore, fails to attribute importance to the very features that drive a confident prediction.

To build a robust contrastive attribution, we must therefore preserve the principle of competition while avoiding this self-canceling behavior. Our core idea is to move the contrastive mechanism from the model's output layer, which operates on saturated probabilities, directly into the attribution maps themselves, computed for the logits. We accomplish this by staging a "local" competition at each pixel to determine which class has the dominant claim on that feature.

Formally, let $H_c = \nabla_x z_c$ denote the base attribution for class $c$. For a chosen set of classes $C' \subseteq C$ of size $K = |C'|$, we compute the standard attribution map $H_c$ for each class $c \in C'$. We then apply a softmax to each input feature, most typically at each spatial location $(i, j)$:

$$\hat{H}_k[i, j] = \frac{\exp(H_k[i, j]/t)}{\sum_{k' \in C'} \exp(H_{k'}[i, j]/t)} \tag{2}$$

where $t$ is a temperature to amplify the contrast. This yields a distribution of attribution over classes at each input feature $\sum_{k \in C'} \hat{H}_k[i, j] = 1$. We denote $q_k(i, j) := \hat{H}_k[i, j]$ for readability. These local class probabilities express how dominant each class is in each spatial location. One might now attempt to directly mimic Eq. (1) by replacing the global softmax weights $p_k$ with the local $q_k(i, j)$, resulting in an attribution of the form:

$$H_c[i, j] - \sum_k q_k(i, j) H_k[i, j].$$

However, this naïve substitution reintroduces the vanishing behavior in another way. The sum includes the self-term $q_c(i, j) H_c[i, j]$, so when class $c$ dominates a pixel, i.e., $q_c(i, j) \approx 1$, the full expression again tends to zero. Summing only over $k \neq c$ is also problematic, as $q_k$ and $H_k$ are strongly correlated, which may lead to overshooting.

Instead, we seek to reduce attribution in proportion to how strongly other classes (not the target class) claim a pixel. Rather than modifying all class gradients, we subtract the share of the target class's own attribution that is explained by its rivals. This leads to the following contrastive form:

$$H_c[i, j] - \sum_{k \neq c} q_k(i, j) \, H_c[i, j] = H_c[i, j] \left( 1 - \sum_{k \neq c} q_k(i, j) \right) = H_c[i, j] \cdot q_c(i, j).$$

Thus, each pixel's attribution is scaled down in proportion to how strongly rival classes explain it, while dominant pixels for class $c$ are preserved.

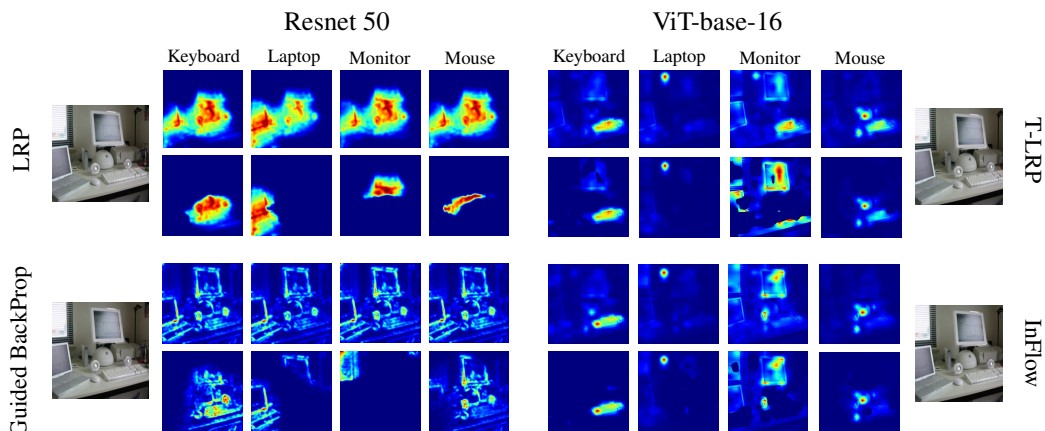

Figure 3: *Contrastive attributions across architectures and methods.* For each baseline (top), our refinement (bottom) sharpens class-specific regions (keyboard, laptop, monitor, mouse). In ResNet-50 the effect is strongest, revealing clear class-specific signals often assumed absent. In ViT-base-16, attributions already cover relevant areas but remain diffuse; our method reduces this blur and highlights the important regions more cleanly.

### 3.3 From Gradients to General Attributions

Most attribution methods can be understood as functions of the gradient: they either use it directly (Input×Gradient), pool it spatially (Grad-CAM), or transform it through propagation rules (Guided Backpropagation, DeepLIFT, etc.). One way to extend our derivation would be to modify each method individually and insert the contrastive reweighting at the gradient level. However, such an approach would be cumbersome and method-specific.

We instead propose a plug-and-play refinement that operates directly on attribution maps, which we call **VAR**, standing for **V**isualizing **A**ctually **R**elevant Features. For a subset $C'$ of classes, we compute the class-wise attributions $\mathcal{H}(x, S, c)$ and normalize them at each pixel using the local spatial softmax of Eq. (2). We denote the resulting distribution by

$$q_c[i,j] := \hat{\mathcal{H}}(x, S, c)[i,j] \,,$$

which expresses the relative dominance of class $c$ at location $(i, j)$. To increase robustness, we average $q_c[i, j]$ over multiple temperatures $t$, producing smoother distributions that capture contrast at different granularities. We then refine the attribution of the target class $c^*$ as

$$\mathcal{H}'(x, S, c^*) = \mathcal{H}(x, S, c^*) \odot q_{c^*} \odot \mathbb{1}_{q_{c^*} - \frac{1}{K} > 0} \,,$$

where $\odot$ denotes element-wise multiplication. Pixels where the target class attribution is near random chance ($q_{c^*} \approx 1/K$) are suppressed, preventing ambiguous regions from diluting the signal. By keeping only $q_{c^*} > 1/K$, we focus on features where $c^*$ clearly dominates. This refinement provides any attribution method with the ability to detect contrastive attributions. In practice, the averaging across temperatures $t \in \{1, 5, 100\}$ stabilizes the competition between classes and removing the requirement to tune this hyperparameter, while the thresholding enforces discriminativeness. The resulting maps $\mathcal{H}'$ thus highlight features that are relevant for the target class $c^*$.

Having defined our class-relevant attribution operator $\mathcal{H}'$, an important consideration is the selection of the Having defined $\mathcal{H}'$, an important consideration is the selection of the set of classes $K$ used for calculation. Predefined sets are natural when the task structure already specifies meaningful contrasts, such as quadrants in location-based metrics or disease subtypes in medical imaging. Model-driven alternatives, such as selecting the top predicted classes or contrasting best with worst, highlight the evidence that separates or defines extreme predictions. In our experiments we adopt predefined sets for location metrics, and the top most probable classes for insertion and randomization tests. Further details and examples are provided in the appendix.

| | Method | Quad-ImageNet | | | Part-Quad-ImageNet | | | COCO | | |
|---|---|---|---|---|---|---|---|---|---|---|
| | | RA | IoU | F1 | RA | IoU | F1 | RA | IoU | F1 |
| Resnet50 | GradCam | 0.88+25% | 0.67+64% | 0.79+38% | 0.31+28% | 0.24+112% | 0.36+87% | 0.18+19% | 0.11+16% | 0.17+12% |
| | GBP | 0.86+144% | 0.26+32% | 0.41+25% | 0.44+146% | 0.08+43% | 0.14+38% | 0.19+30% | 0.09+3% | 0.15+2% |
| | Guide-GC | 0.91+21% | 0.34+31% | 0.50+23% | 0.50+24% | 0.12+49% | 0.21+42% | 0.23+16% | 0.10+8% | 0.16+8% |
| | IxG | 0.55+37% | 0.20+0% | 0.33+0% | 0.25+47% | 0.06+0% | 0.11+0% | 0.13+11% | 0.09+0% | 0.15+0% |
| | IG | 0.56+36% | 0.20+0% | 0.34+0% | 0.28+48% | 0.06+0% | 0.12+0% | 0.14+11% | 0.09+0% | 0.15+0% |
| | LRP | 0.88+56% | 0.69+97% | 0.79+55% | 0.37+49% | 0.22+117% | 0.34+90% | 0.21+20% | 0.13+8% | 0.20+7% |
| | Avg. Improvement | +53.17% | +37.33% | +23.50% | +57.00% | +53.50% | +42.83% | +17.83% | +5.83% | +4.83% |
| ViT-base-16 | Bi-attn | 0.94+31% | 0.71+180% | 0.82+103% | 0.51+40% | 0.28+309% | 0.40+222% | 0.30+43% | 0.16+52% | 0.23+42% |
| | GradCam | 0.91+6% | 0.62+16% | 0.75+10% | 0.58+11% | 0.27+39% | 0.39+32% | 0.31+10% | 0.15+11% | 0.22+9% |
| | InFlow | 0.86+21% | 0.56+126% | 0.71+78% | 0.53+23% | 0.20+198% | 0.31+153% | 0.29+20% | 0.13+23% | 0.20+21% |
| | Grad-Rollout | 0.73+76% | 0.53+113% | 0.68+71% | 0.40+94% | 0.20+197% | 0.30+148% | 0.24+30% | 0.12+19% | 0.19+17% |
| | T-attn | 0.93+32% | 0.71+180% | 0.82+102% | 0.47+38% | 0.29+321% | 0.40+229% | 0.29+44% | 0.16+53% | 0.23+43% |
| | T-LRP | 0.77+35% | 0.51+105% | 0.66+65% | 0.47+36% | 0.20+201% | 0.31+152% | 0.27+17% | 0.12+20% | 0.19+18% |
| | Gradient | 0.93+4% | 0.57+3% | 0.70+2% | 0.50+8% | 0.34+11% | 0.47+9% | 0.30+10% | 0.17+2% | 0.25+2% |
| | Avg. Improvement | +29.29% | +103.29% | +61.57% | +35.71% | +182.29% | +135.00% | +26.29% | +28.57% | +25.71% |

Table 1: *Consistent improvement of attributions.* Across 11 different attribution methods considering convolutional and transformer based architectures, quantitative metrics measured using Region Attribution (RA), Intersection over Union (IoU), and F1 get consistently improved by a wide margin. We provide results for more architectures in App. Tab. 1 showing similar trends. We show the value that the method achieves when augmented with VAR and in percent the level of improvement.

## 4 EXPERIMENTS

We empirically evaluate our novel attribution pipeline, which we call VAR, with 11 common attribution scores on three benchmark settings: ability to localize, insertion tests, and randomization-based sanity checks. To assess localization ability, we consider the validation set of ImageNet Russakovsky et al. (2015), MS-COCO Lin et al. (2014), and the Grid Pointing Game on ImageNet Rao et al. (2022a). We assess the quality of attributions by measuring how well these match annotated bounding boxes and segmentation masks. For insertion tests, we quantitatively evaluate attributions using standard perturbation testing, which measures the importance of pixels. We employ the insertion method following the approaches of XRAI Kapishnikov et al. (2019), which allows for systematic evaluation of how the addition of information impacts model confidence. Given the computational complexity, we use the first 1k. To validate robustness, we conduct sanity checks using randomization tests on 10k images on ImageNet Adebayo et al. (2018).

We consider different architectures, including ResNet-50 He et al. (2016), Vision Transformer B/16 (ViT) Dosovitskiy et al. (2020), and provide further results for DenseNet-121 Huang et al. (2017), Wide ResNet-50-2 Zagoruyko & Komodakis (2016), and ConvNeXt Liu et al. (2022) in the Appendix. All models are ImageNet-pretrained (from PyTorch) and used in standard classification mode.For attribution, we adopt widely used methods for CNNs Grad-CAM, Guided Backprop, IG, IxG, Guided Grad-CAM Selvaraju et al. (2017); Springenberg et al. (2015); Sundararajan et al. (2017); Shrikumar et al. (2017); Kokhlikyan et al. (2020) and for ViTs Grad-CAM, InFlow, Grad-Rollout, Bi-Attn, T-attn, T-LRP, gradient saliency Walker et al. (2025a); Abnar & Zuidema (2020); Chen et al. (2022); Yuan et al. (2021a); Chefer et al. (2021b). Because transformer saliency maps are blurry, we multiply them with the input (similar to IxG) for illustration..

### 4.1 LOCALIZATION ABILITY

**Metrics** We assess attribution quality by measuring how well attributions align with the actual object regions. The Region Attribution (RA) metric quantifies what portion of the total attribution weight falls within the target region, providing insight into attribution focus. The Intersection over Union (IoU) measures the spatial overlap between the attribution map and the ground truth region, and the F1-score score is computed between attributed and true pixels of the target object. Before evaluation, to prevent methods from being unduly rewarded for producing diffuse attributions, we apply a Gaussian blur to the attribution maps and ensure a fair comparison across different approaches following Rao et al. (2022b). For both setups we use the target classes for $\mathcal{C}'$.

**Grid Pointing** For the grid-pointing game, we compile a $2 \times 2$ grid of random images from ImageNet validation set, which we call Quad-ImageNet resulting in 12500 images. Across attribution scores and architectures, we observe that VAR never degrades performance, at minimum we see that for specific methods and benchmark setups it is on par with the standard pipeline. Most of the time,

Integrated Gradients      Guided Backprop      Input×Gradient

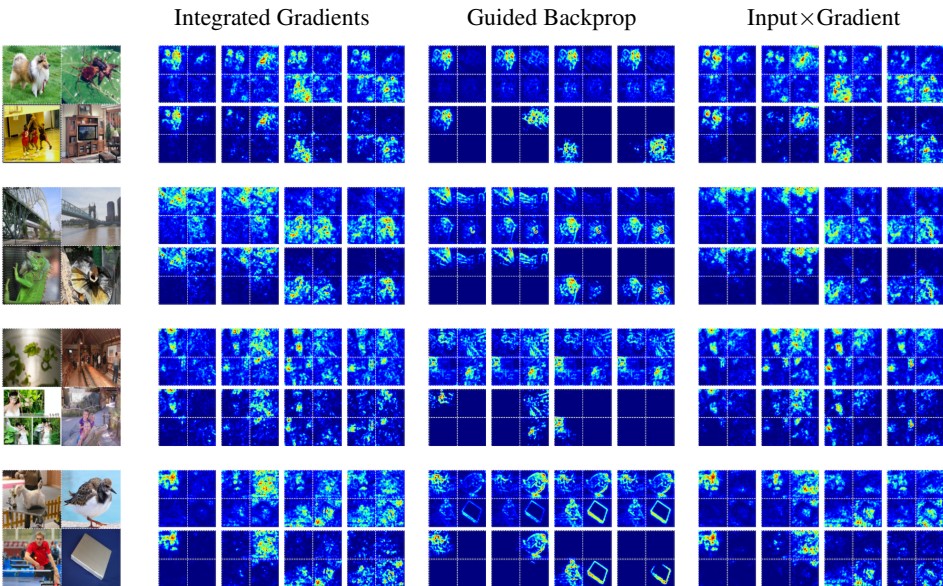

Figure 4: *VAR on the Grid Pointing Game.* We show examples from the grid pointing game for methods most affected by our framework (as columns: Integrated Gradient, Guided Backpropagation, Input×Gradient) for ResNet50. Input Images are given on the left, for each we provide vanilla attribution methods (top rows) and augmented with VAR (bottom rows). For each, we show the attribution for the four different classes in the grid as columns.

we see that the **VAR pipeline greatly improves localization ability** (see Tab. 1). For ResNet50, we observe substantial gains in RA, IoU, and F1, with an average improvement of upto $+53\%$, $+37\%$ and $+42\%$ across different attribution scores. For specific methods, such as Guided Backpropagation, the RA score even doubles on the Quad-Imagenet benchmark. For the ViT model, IoU and F1 scores more than double. This strong improvement can be partially attributed to VAR filtering out uniformly unimportant regions in the noisy attribution maps produced for ViT.

Qualitatively, we also observe these improvements, now capturing both the *distinguishing* as well as *common* features of closely related classes (cf. Fig. 4). These results show that the VAR pipeline enhances attribution methods in precisely localizing features most relevant to the classification.

**MS-COCO**   For MS-COCO, we use the whole validation set. We filter objects that are smaller than 1% of the image and objects for which the model has a confidence less than $10^{-4}$. We observe similar trends, albeit more modest than on Quad-Imagenet, achieving an average improvement of $+17.83\%$, $+5.83\%$, and $+4.83\%$ on RA, IoU, and F1 respectively. Again we observe that through VAR, the attributions focus on more discriminative features rather than entire object regions. COCO's natural images contain multiple objects with complex backgrounds, making precise localization more challenging, yet with VAR we do improve F1 scores across regardless of attribution scoring approach on both ResNet50 and ViT, indicating better overall localization despite the more challenging context. We provide results for different convolutional and transformer-based architectures in App. Tab. 1 showing similar improvements through VAR.

## 4.2 INSERTION ABLATIONS

We follow the benchmark evaluation scheme for saliency maps of Kapishnikov et al. (2019) measuring how effectively an attribution method identifies the relevant image regions for a model's decision. We use the Performance Information Curve (PIC) framework, where we start with a blurred image, progressively restoring high-attribution pixels, measuring model confidence, and when the model returns to its initial prediction. This process produces Performance Information Curves that track how classification performance evolves as information is reintroduced. To quantify overall perfor-

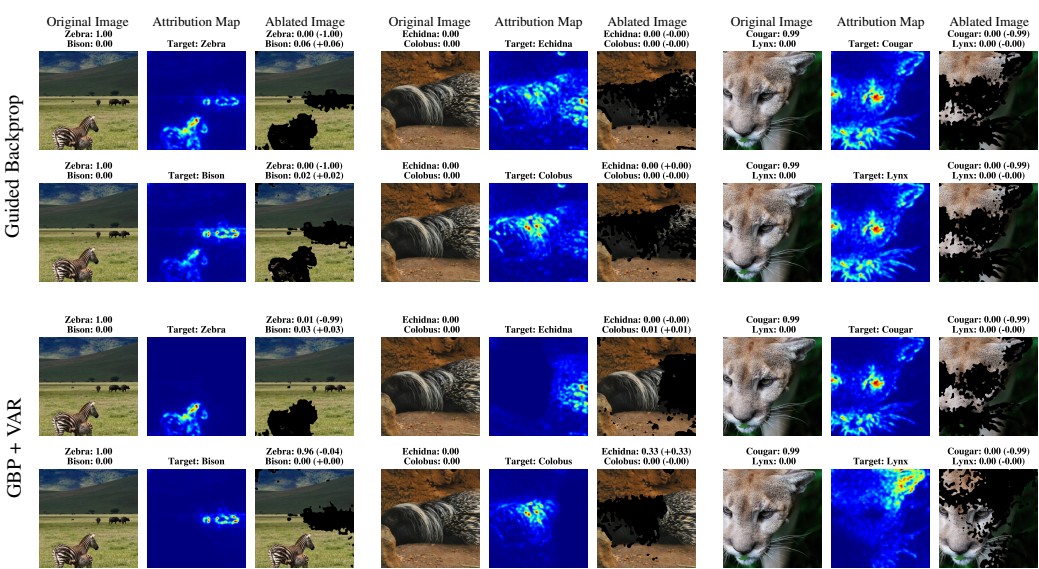

Figure 5: *Qualitative example of the ablation study.* For GBP (top) and GBP with VAR (bottom) we provide examples from the insertion/deletion ablation. For each, we show the original image with class softmax scores for two classes, the attribution map for each of the classes, and the attribution-based intervention mask on each of the classes with resulting changes in class softmax scores.

| Method | ResNet50 | WideResNet50-2 | DenseNet121 | ConvNeXT |
|---|---|---|---|---|
| IG | 0.49+0% | 0.53+2% | 0.48+2% | 0.37+0% |
| GBP | 0.57+2% | 0.60+2% | 0.53+2% | 0.34−3% |
| IxG | 0.46+2% | 0.49+2% | 0.43+2% | 0.35+0% |
| Guide-GC | 0.58+0% | 0.61+0% | 0.53+0% | 0.51+0% |
| GradCam | 0.61+11% | 0.63+9% | 0.49+2% | 0.50−2% |
| LRP | 0.66+0% | 0.71+1% | 0.41+0% | - |
| Avg. Improvement | +2.50% | +2.67% | +1.33% | −1% |

(a) CNN-based architectures

| Method | ViT-base-8 | ViT-base-16 | ViT-base-32 |
|---|---|---|---|
| Bi-attn | 0.78+8% | 0.75+9% | 0.65+8% |
| T-attn | 0.75+10% | 0.74+9% | 0.65+14% |
| InFlow | 0.79+11% | 0.79+14% | 0.66+12% |
| Gradient | 0.75−1% | 0.72−3% | 0.67+3% |
| Grad-Rl | 0.76+10% | 0.76+10% | 0.66+16% |
| T-LRP | 0.74+12% | 0.75+10% | 0.67+14% |
| Avg. Improvement | +8.33% | +8.17% | +11.17% |

(b) Transformer-based architectures

Table 2: *Improving transformer attributions on insertion ablations.* Our proposed pipeline provides on par performance in terms of AIC for convolutional architectures and improve almost all transformer attribution methods by 10% across architectures.

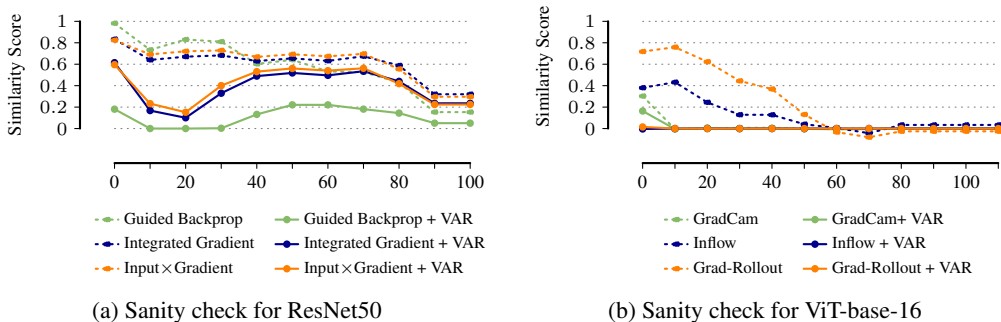

(a) Sanity check for ResNet50      (b) Sanity check for ViT-base-16

Figure 6: *Sanity check by network randomization.* We show similarity between attributions before and after randomization of x% of network layers for standard attribution (dashed) and when augmented with VAR (solid). **Lower is better.** Randomization is from back to front of the network following the strategy of Adebayo et al. (2018).

mance, we report the Area under Accuracy Information Curve (AIC), which summarizes accuracy across different information levels. We evaluate on ImageNet and report results in Table 2. We find that our pipeline keeps performance on par with the vanilla pipeline for convolutional architectures, and greatly **improves performance for transformer-based architectures across a wide range of transformer-specific attribution methods**.

### 4.3 SANITY CHECKS

To verify that attribution methods reflect the representation the model learned, we conduct cascading randomization tests following Adebayo et al. (2018). These tests progressively randomize model parameters from output to input layers, measuring how attribution maps change as model knowledge is systematically destroyed. We follow the same procedure as in the original paper. We measure the Spearman correlation (Fig. 6) between attributions before and after randomization and additionally provide cosine similarity and Pearson correlation across different architectures in App. Figure 8–14, which show similar trends. As the later parts of the network is randomized, ideally there is little information left about the target in the attribution maps. However, as discussed by Binder et al. (2023), this randomization-based approach has shortcomings as it "preserves scales of forward pass activations with high probability". Hence, we are primarily interested in the *relative change* between attributions with and without VAR.

We find that attribution maps using the VAR pipeline yield better results for all baseline methods and across randomization percentages. For Guided-Backprop and Input × Gradient, the improvement is most pronounced, as well as for randomizing the latest layers, which carry most of the conceptual meaning for the classification. Intriguingly, for ViT models, we observe that after randomization at any point in the network the similarity score is virtually zero, meaning that specific **attribution methods correctly taking class contrast into account can pass the sanity check**.

## 5 DISCUSSION & CONCLUSION

Attribution methods are widely employed, yet also critically discussed for seemingly not faithfully describing the classification-relevant features in the input. In this work, we reconsidered the common paradigm of computing attributions on a target logit. We found that this does not adequately reflect the actual decision-making process of a classification model and instead propose to compute distributions of attributions across multiple classes. This change led to drastically improved attributions that reflect object- and instance-specific features, highlight class-discriminative as well as shared features, which attributions on logits in isolation cannot provide. Most importantly, we demonstrate that even the most common attribution methods–especially for CNNs–already encode rich class-specific information, but this signal has remained largely hidden by vanishing gradients.

To quantitatively substantiate these claims, we provided extensive evaluation across attribution methods, convolutional as well transformer-based architectures, and different benchmarks for saliency maps, including the grid pointing game Rao et al. (2022a), sanity checks for saliency maps Adebayo et al. (2018), and insertion tests Kapishnikov et al. (2019). Our evaluation of interpretability is by no means exhaustive, as a wide array of different benchmarks has been proposed over time. We here focused on the most common evaluation protocols in the literature.

The reconsideration of where and how to apply attributions is method and model agnostic, training-free, and faithful to the target model in that we do not use surrogates or other, eg. generative, models that could introduce new biases. Interestingly, Gairola et al. (2025) recently found that training with binary cross-entropy loss significantly improves attributions in terms of downstream benchmarks, arguing for BCE for improved post-hoc explanations. Our findings provide a reason why this is the case, as BCE incentivizes the network to learn class-specific features, which will consequently appear in attributions even in the standard attribution pipeline looking at a logit in isolation. Here, we compute attributions as distributions over classes, which is training-free, thus maintaining the accuracy of the model, and not only reveals class-specific but also features shared across classes, which BCE discourages in training, and reveals object-specific attributions in multi-object settings.

We anticipate that this pipeline supercharges existing attribution methods to *understand the distinguishing features a model uses for prediction* and is a versatile tool that can be combined with any current and future attribution method independent of target architecture or attribution type.

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

# A METHOD

## A.1 SELECTING THE SET OF CLASSES

Having defined our class-relevant attribution operator $\mathcal{C}_{\mathcal{H}}$, an important consideration is the selection of the set of classes $K$ used for calculation. We explore three approaches for class selection, each offering distinct advantages depending on the specific analysis goals and application context.

**Predefined Class Sets.** The canonical approach is to use a predefined set of classes $K$ that are of particular interest. This is especially useful in contexts where specific class comparisons have natural interpretations. For example, in a grid-pointing game where users must identify the quadrant containing a particular object, the four quadrant classes directly correspond to the task structure. Similarly, in medical applications, contrasting disease subtypes can highlight discriminative features that aid differential diagnosis. This approach ensures that the resulting attributions focus on distinctions that are meaningful to the particular application domain. However, this approach requires specific knowledge about the task, which is often not available. The following approaches are data- and model-driven and, hence, do not require prior knowledge to select classes.

**Top-$k$ Most Probable Classes.** A model dependent approach to class selection involves choosing the $k$ classes with highest predicted probabilities and the class with the lowest probability for a given input. This approach is particularly effective for highlighting the features that distinguish between the most plausible classifications for a given input, but also reveal information that is shared between highly related classes that are likely among the highest probabilities. As these classes represent the top candidates for the final classification, contrasting their attribution maps reveals the most decision-relevant features.

**Best–vs–Worst Classes.** The third approach compares the highest-probability class against the lowest-probability class: $K = \{c_{\max}, c_{\min}\}$ where $c_{\max} = \arg\max_c S_c(x)$ and $c_{\min} = \arg\min_c S_c(x)$. Such extreme can surprisingly reveal the most distinctive characteristics of the input as interpreted by the model, by showing which features are most critical for pushing the model toward or away from certain classifications.

# B EVALUATION METRICS

In our experimental setup, we evaluate attribution methods across several metrics to assess their efficacy in highlighting relevant features for model predictions. We define an input as a vector $x \in \mathbb{R}^d$, and a model as a function $S : \mathbb{R}^d \to \mathbb{R}^C$, where $C$ is the number of classes in the classification problem. The final classification is performed via an argmax over $S(x)$. An explanation method provides an explanation map $\mathcal{H} : \mathbb{R}^d \times S \times \{1, ..., C\} \to \mathbb{R}^d$ that maps an input, a model, and optionally a target class to an attribution map of the same shape as the input.

## B.1 LOCALIZATION METRICS

We evaluate attribution methods using two datasets: a Grid Pointing Game based on ImageNet and COCO dataset with segmentation masks. For both evaluations, we apply the same set of metrics, treating both bounding boxes and segmentation masks as regions of interest $R$ in the image. We match the region of interest with the correct attribution map $\mathcal{H}_c$ i.e. for the first quadrant we also take the first attribution map. We only take the positive part of $\mathcal{H}_c$. Before evaluation, we apply a Gaussian blur with a kernel size of $11 \times 11$ to the attribution maps

$$\tilde{\mathcal{H}}_c = \mathcal{G}_\sigma * \mathcal{H}_c \,,$$

where $\mathcal{G}_\sigma$ is a Gaussian kernel with standard deviation $\sigma$ and $*$ denotes the convolution operation. This preprocessing is common to prevent methods from being unduly rewarded for producing diffuse attribution maps. We then compute the following metrics:

### B.1.1 REGION ATTRIBUTION

We quantify what fraction of the total positive attribution falls within the region of interest by

$$\text{RA} = \frac{\sum_{i \in R} \tilde{\mathcal{H}}_c(i)}{\sum_i \tilde{\mathcal{H}}_c(i)} \,.$$

### B.1.2 INTERSECTION OVER UNION (IoU)

We compute the overlap between the attribution map and the region of interest as

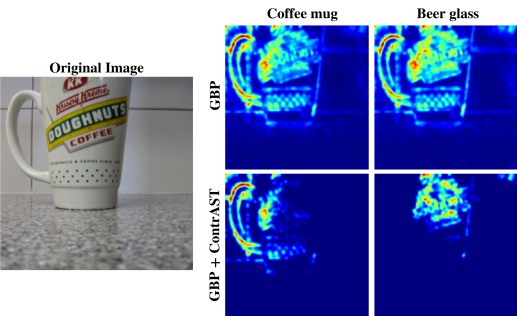

Figure 7: VAR highlights discriminative features. Standard Guided Backpropagation (GBP) produces nearly identical attributions for both "coffee mug" and "beer glass" classes (top row), while our approach (bottom row) clearly emphasizes the distinguishing features of each class—the handle for coffee mug and the cup shape for beer glass. This focus on discriminative features aids interpretability but can result in lower insertion test scores which reward highlighting the entire object.

$$\text{IoU} = \frac{|(\tilde{\mathcal{H}}_c \cap R|}{|\tilde{\mathcal{H}}_c \cup R|}.$$

### B.1.3 PRECISION AND RECALL

To calculate precision and recall, we use the commen intersection-based formulas

$$\text{Precision} = \frac{|\tilde{\mathcal{H}}_c \cap R|}{|\tilde{\mathcal{H}}_c|}, \quad \text{Recall} = \frac{|\tilde{\mathcal{H}}_c \cap R|}{|R|}.$$

### B.1.4 F1 SCORE

To calculate F1, we make use of the previously defined precision and recall metrics, caluclating

$$\text{F1} = \frac{2 \cdot \text{Precision} \cdot \text{Recall}}{\text{Precision} + \text{Recall}}.$$

## C LLM USE

In this work, we used GPT-5 for both writing and coding support. On the writing side, it assisted with editing and condensing text to improve clarity. For coding, GPT-5 was used for debugging, providing autocomplete suggestions in VS Code, and generating code for LaTeX figures.

## D ADDITIONAL RESULST

### D.1 LOCALIZATION

We provide additional results for all the architectures mentioned in the Experiments in Table 3. The trend remains the same across architectures and methods; if they are augmented using VAR they improve the localization metrics and trade-off recall. Additionally we provide plots similar to Figure 4 for all these architectures in Figure 15-21.

### D.2 SANITY CHECKS

We show the sanity check plots for these additional architectures in Figure 8-14.

| | Method | Quad-ImageNet | | | Part-Quad-ImageNet | | | COCO | | |
|---|---|---|---|---|---|---|---|---|---|---|
| | | RA | IoU | F1 | RA | IoU | F1 | RA | IoU | F1 |
| Resnet50 | GradCam | 0.88+25% | 0.67+64% | 0.79+38% | 0.31+28% | 0.24+112% | 0.36+87% | 0.18+19% | 0.11+16% | 0.17+12% |
| | GBP | 0.86+144% | 0.26+32% | 0.41+25% | 0.44+146% | 0.08+43% | 0.14+38% | 0.19+30% | 0.09+3% | 0.15+2% |
| | Guide-GC | 0.91+21% | 0.34+31% | 0.50+23% | 0.50+24% | 0.12+49% | 0.21+42% | 0.23+16% | 0.10+8% | 0.16+8% |
| | IxG | 0.55+37% | 0.20+0% | 0.33+0% | 0.25+47% | 0.06+0% | 0.11+0% | 0.13+11% | 0.09+0% | 0.15+0% |
| | IG | 0.56+36% | 0.20+0% | 0.34+0% | 0.28+48% | 0.06+0% | 0.12+0% | 0.14+11% | 0.09+0% | 0.15+0% |
| | LRP | 0.88+56% | 0.69+97% | 0.79+55% | 0.37+49% | 0.22+117% | 0.34+90% | 0.21+20% | 0.13+8% | 0.20+7% |
| | Avg. Improvement | +53.17% | +44.8% | +28.2% | +57.0% | +80.25% | +64.25% | +18.33% | +6.17% | +5.33% |
| Wide-Resnet502 | GradCam | 0.88+22% | 0.66+62% | 0.78+37% | 0.30+23% | 0.24+108% | 0.36+84% | 0.18+14% | 0.11+11% | 0.17+9% |
| | GBP | 0.89+109% | 0.28+42% | 0.43+32% | 0.47+108% | 0.09+57% | 0.16+51% | 0.20+24% | 0.10+2% | 0.16+2% |
| | Guide-GC | 0.92+17% | 0.35+32% | 0.51+24% | 0.51+20% | 0.13+53% | 0.22+46% | 0.24+13% | 0.10+6% | 0.17+6% |
| | IxG | 0.62+38% | 0.20+0% | 0.33+0% | 0.27+48% | 0.06+0% | 0.11+0% | 0.15+12% | 0.10+0% | 0.15+0% |
| | IG | 0.62+37% | 0.20+0% | 0.34+0% | 0.31+48% | 0.06+0% | 0.12+0% | 0.15+12% | 0.10+0% | 0.15+0% |
| | LRP | 0.89+49% | 0.72+115% | 0.82+65% | 0.37+46% | 0.22+132% | 0.34+102% | 0.22+22% | 0.13+11% | 0.20+9% |
| | Avg. Improvement | +45.33% | +41.83% | +26.33% | +48.83% | +58.33% | +47.17% | +16.17% | +5.0% | +4.33% |
| Densenet121 | GradCam | 0.60+17% | 0.37+6% | 0.48−2% | 0.22+30% | 0.15+51% | 0.23+39% | 0.11−11% | 0.07−25% | 0.11−25% |
| | GBP | 0.85+158% | 0.25+27% | 0.40+21% | 0.41+159% | 0.08+35% | 0.14+32% | 0.19+36% | 0.10+3% | 0.15+3% |
| | Guide-GC | 0.71+28% | 0.26+12% | 0.40+9% | 0.37+34% | 0.10+30% | 0.17+26% | 0.17+3% | 0.08−5% | 0.14−5% |
| | IxG | 0.46+31% | 0.20+0% | 0.33+0% | 0.20+42% | 0.06+0% | 0.11+0% | 0.13+10% | 0.09+0% | 0.15+0% |
| | IG | 0.50+34% | 0.20+0% | 0.34+0% | 0.24+46% | 0.06+0% | 0.12+0% | 0.14+11% | 0.09+0% | 0.15+0% |
| | LRP | 0.44+24% | 0.25+0% | 0.40+0% | 0.19+35% | 0.07+0% | 0.12+0% | 0.15+5% | 0.12+0% | 0.18+0% |
| | Avg. Improvement | +48.67% | +7.5% | +4.67% | +57.67% | +19.33% | +16.17% | +9.0% | −4.5% | −4.5% |
| Convnext | GradCam | 0.96+2% | 0.55−7% | 0.70−6% | 0.48+8% | 0.29+31% | 0.42+24% | 0.28+8% | 0.15+2% | 0.23+2% |
| | GBP | 0.52+26% | 0.20+0% | 0.33+0% | 0.19+33% | 0.06+0% | 0.11+0% | 0.15+13% | 0.09+0% | 0.15+0% |
| | Guide-GC | 0.96+1% | 0.35+1% | 0.52+1% | 0.58+5% | 0.16+2% | 0.26+2% | 0.31+5% | 0.14+1% | 0.22+1% |
| | IxG | 0.51+27% | 0.20+0% | 0.33+0% | 0.19+33% | 0.06+0% | 0.11+0% | 0.15+13% | 0.09+0% | 0.15+0% |
| | IG | 0.64+35% | 0.21+0% | 0.34+0% | 0.26+49% | 0.06+1% | 0.12+0% | 0.15+16% | 0.09+0% | 0.15+0% |
| | Avg. Improvement | +18.20% | −1.20% | −1% | +25.60% | +6.80% | +5.40% | +11.00% | +0.60% | +0.60% |
| ViT-base-8 | Bi-attn | 0.91+48% | 0.62+149% | 0.76+89% | 0.56+61% | 0.25+272% | 0.36+199% | 0.29+45% | 0.14+32% | 0.21+27% |
| | GradCam | 0.83+8% | 0.49+18% | 0.64+12% | 0.61+11% | 0.28+46% | 0.40+36% | 0.30+13% | 0.14+9% | 0.21+7% |
| | InFlow | 0.82+18% | 0.47+89% | 0.63+58% | 0.59+19% | 0.18+165% | 0.28+131% | 0.32+18% | 0.12+14% | 0.19+13% |
| | Grad-Rollout | 0.71+51% | 0.45+80% | 0.61+53% | 0.48+60% | 0.20+197% | 0.30+147% | 0.26+27% | 0.12+14% | 0.19+12% |
| | T-attn | 0.90+53% | 0.63+152% | 0.76+90% | 0.51+76% | 0.28+322% | 0.40+230% | 0.28+56% | 0.14+34% | 0.22+29% |
| | LRP | 0.76+25% | 0.42+69% | 0.58+46% | 0.54+24% | 0.20+195% | 0.30+148% | 0.28+16% | 0.12+14% | 0.19+13% |
| | Gradient | 0.90+7% | 0.49+7% | 0.64+5% | 0.57+11% | 0.35+20% | 0.48+16% | 0.31+17% | 0.16+2% | 0.23+1% |
| | Avg. Improvement | +30.0% | +80.57% | +50.43% | +37.43% | +173.86% | +129.57% | +27.43% | +17.0% | +14.57% |
| ViT-base-16 | Bi-attn | 0.94+31% | 0.71+180% | 0.82+103% | 0.51+40% | 0.28+309% | 0.40+222% | 0.30+43% | 0.16+52% | 0.23+42% |
| | GradCam | 0.91+6% | 0.62+16% | 0.75+10% | 0.58+11% | 0.27+39% | 0.39+32% | 0.31+10% | 0.15+11% | 0.22+9% |
| | InFlow | 0.86+21% | 0.56+126% | 0.71+78% | 0.53+23% | 0.20+198% | 0.31+153% | 0.29+20% | 0.13+23% | 0.20+21% |
| | Grad-Rollout | 0.73+76% | 0.53+113% | 0.68+71% | 0.40+94% | 0.20+197% | 0.30+148% | 0.24+30% | 0.12+19% | 0.19+17% |
| | T-attn | 0.93+32% | 0.71+180% | 0.82+102% | 0.47+38% | 0.29+321% | 0.40+229% | 0.29+44% | 0.16+53% | 0.23+43% |
| | LRP | 0.77+35% | 0.51+105% | 0.66+65% | 0.47+36% | 0.20+201% | 0.31+152% | 0.27+17% | 0.12+20% | 0.19+18% |
| | Gradient | 0.93+4% | 0.57+3% | 0.70+2% | 0.50+8% | 0.34+11% | 0.47+9% | 0.30+10% | 0.17+2% | 0.25+2% |
| | Avg. Improvement | +29.29% | +103.29% | +61.57% | +35.71% | +182.29% | +135.0% | +24.86% | +25.71% | +21.71% |
| ViT-base-32 | Bi-attn | 0.86+71% | 0.62+149% | 0.75+87% | 0.36+80% | 0.24+263% | 0.36+195% | 0.21+37% | 0.13+33% | 0.20+27% |
| | GradCam | 0.78+18% | 0.51+50% | 0.65+30% | 0.41+28% | 0.22+119% | 0.32+91% | 0.22+25% | 0.13+15% | 0.20+13% |
| | InFlow | 0.78+21% | 0.56+124% | 0.70+75% | 0.38+24% | 0.19+176% | 0.29+136% | 0.22+19% | 0.12+21% | 0.19+17% |
| | Grad-Rollout | 0.66+91% | 0.51+106% | 0.66+66% | 0.27+112% | 0.17+151% | 0.27+119% | 0.17+28% | 0.11+13% | 0.18+12% |
| | T-attn | 0.84+70% | 0.62+146% | 0.74+86% | 0.35+77% | 0.25+267% | 0.36+197% | 0.20+35% | 0.14+34% | 0.20+27% |
| | LRP | 0.66+25% | 0.46+85% | 0.61+53% | 0.31+51% | 0.17+147% | 0.26+115% | 0.20+16% | 0.11+11% | 0.18+9% |
| | Gradient | 0.79+19% | 0.51+19% | 0.65+12% | 0.36+27% | 0.23+53% | 0.34+42% | 0.21+17% | 0.13+11% | 0.20+8% |
| | Avg. Improvement | +48.43% | +97.0% | +58.43% | +57.0% | +168.0% | +127.86% | +25.29% | +19.71% | +16.14% |

Table 3: *Consistent improvement of attributions.* Across 11 different attribution methods considering convolutional and transformer based architectures, quantitative metrics measured using Region Attribution (RA), Intersection over Union (IoU), and F1 get consistently improved by a wide margin. We provide results for more architectures in App. Tab. 1 showing similar trends.

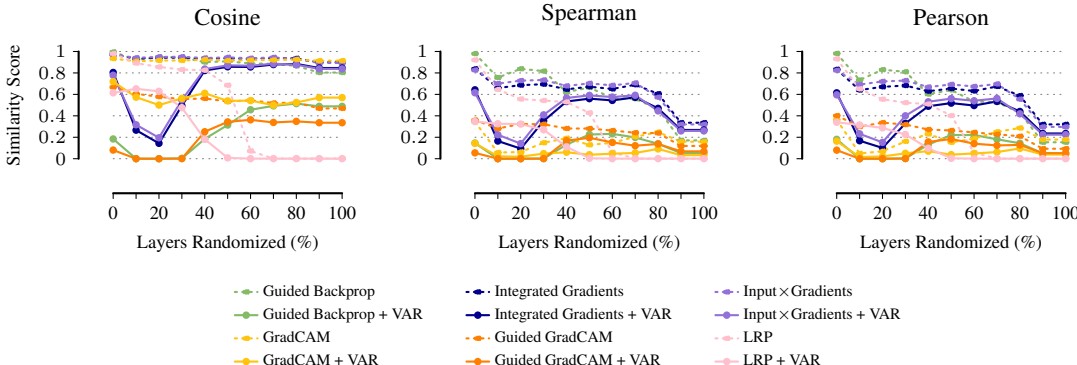

Figure 8: ResNet50: VAR improves all base methods under randomization [Lower is better]. For all methods and for varying level of randomization, we measure the similarity between the attention map for the unperturbed network and the randomized network. Dashed lines are base methods, solid lines when augmenting with VAR, which improve the corresponding baseline method.

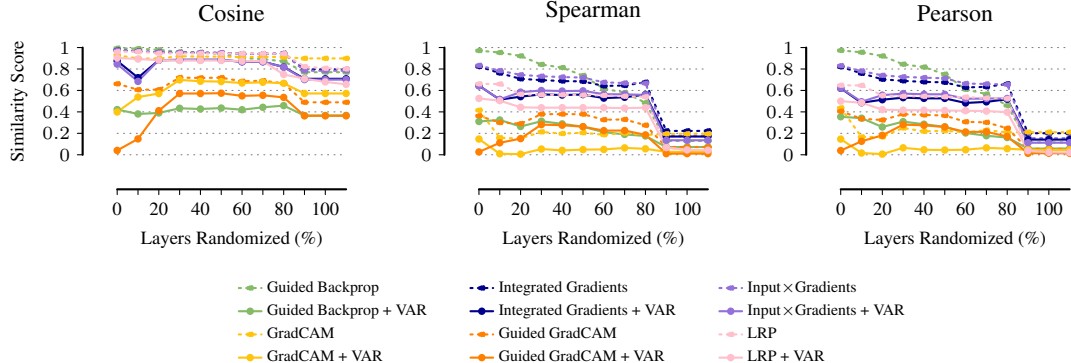

Figure 9: DenseNet121: VAR improves all base methods under randomization [Lower is better]. For all methods and for varying level of randomization, we measure the similarity between the attention map for the unperturbed network and the randomized network. Dashed lines are base methods, solid lines when augmenting with VAR, which always improve the corresponding baseline method.

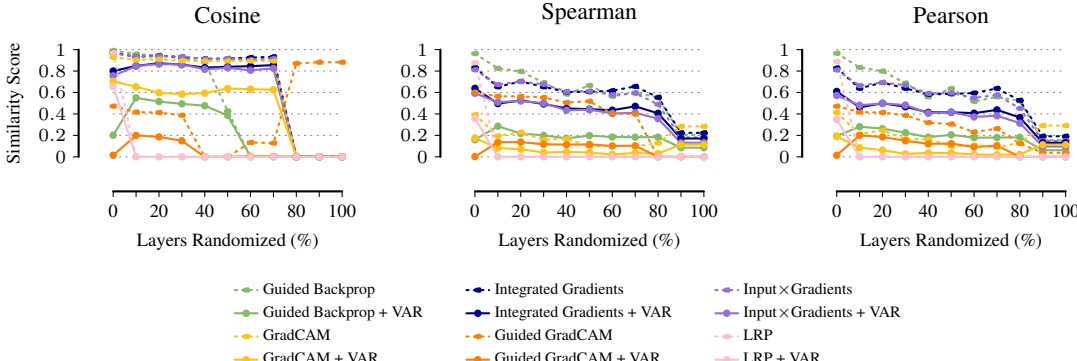

Figure 10: WRN50-2: VAR improves all base methods under randomization [Lower is better]. For all methods and for varying level of randomization, we measure the similarity between the attention map for the unperturbed network and the randomized network. Dashed lines are base methods, solid lines when augmenting with VAR, which always improve the corresponding baseline method.

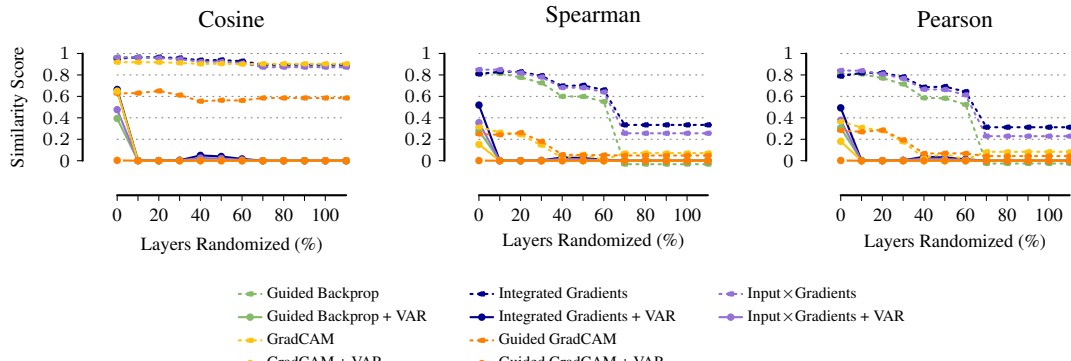

Figure 11: ConvNext: VAR improves all base methods under randomization [Lower is better]. For all methods and for varying level of randomization, we measure the similarity between the attention map for the unperturbed network and the randomized network. Dashed lines are base methods, solid lines when augmenting with VAR, which improve the corresponding baseline method.

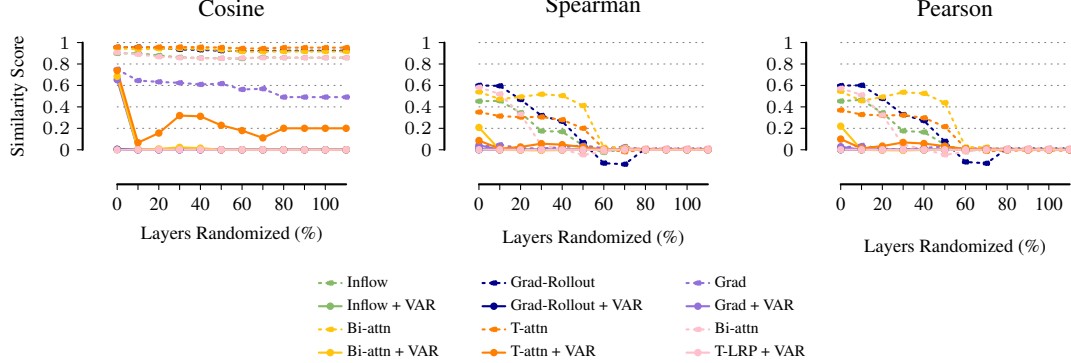

Figure 12: ViT-base-8: VAR improves all base methods under randomization [Lower is better]. For all methods and for varying level of randomization, we measure the similarity between the attention map for the unperturbed network and the randomized network. Dashed lines are base methods, solid lines when augmenting with VAR, which improve the corresponding baseline method.

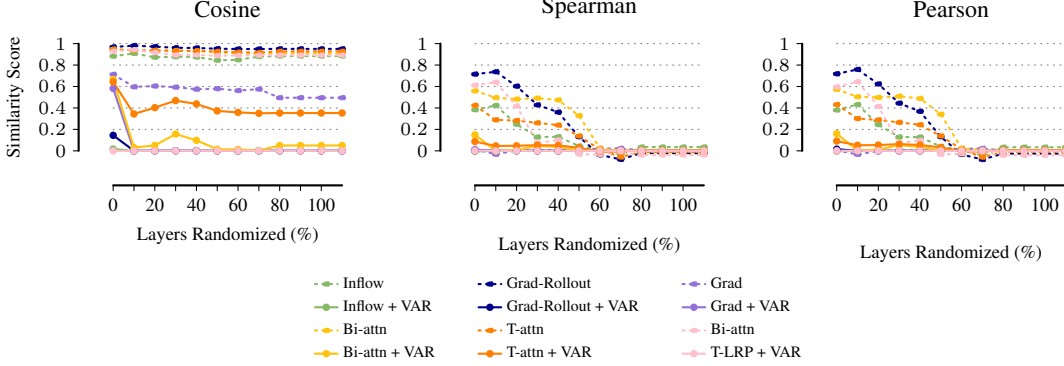

Figure 13: ViT-base-16: VAR improves all base methods under randomization [Lower is better]. For all methods and for varying level of randomization, we measure the similarity between the attention map for the unperturbed network and the randomized network. Dashed lines are base methods, solid lines when augmenting with VAR, which improve the corresponding baseline method.

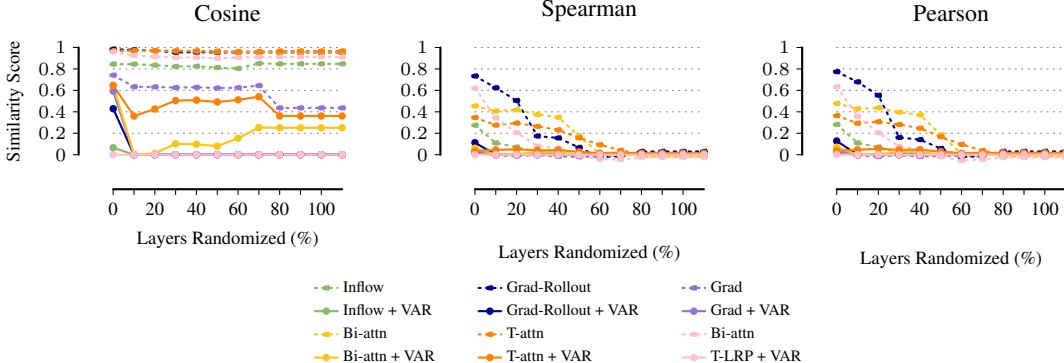

Figure 14: ViT-base-32: VAR improves all base methods under randomization [Lower is better]. For all methods and for varying level of randomization, we measure the similarity between the attention map for the unperturbed network and the randomized network. Dashed lines are base methods, solid lines when augmenting with VAR, which improve the corresponding baseline method.

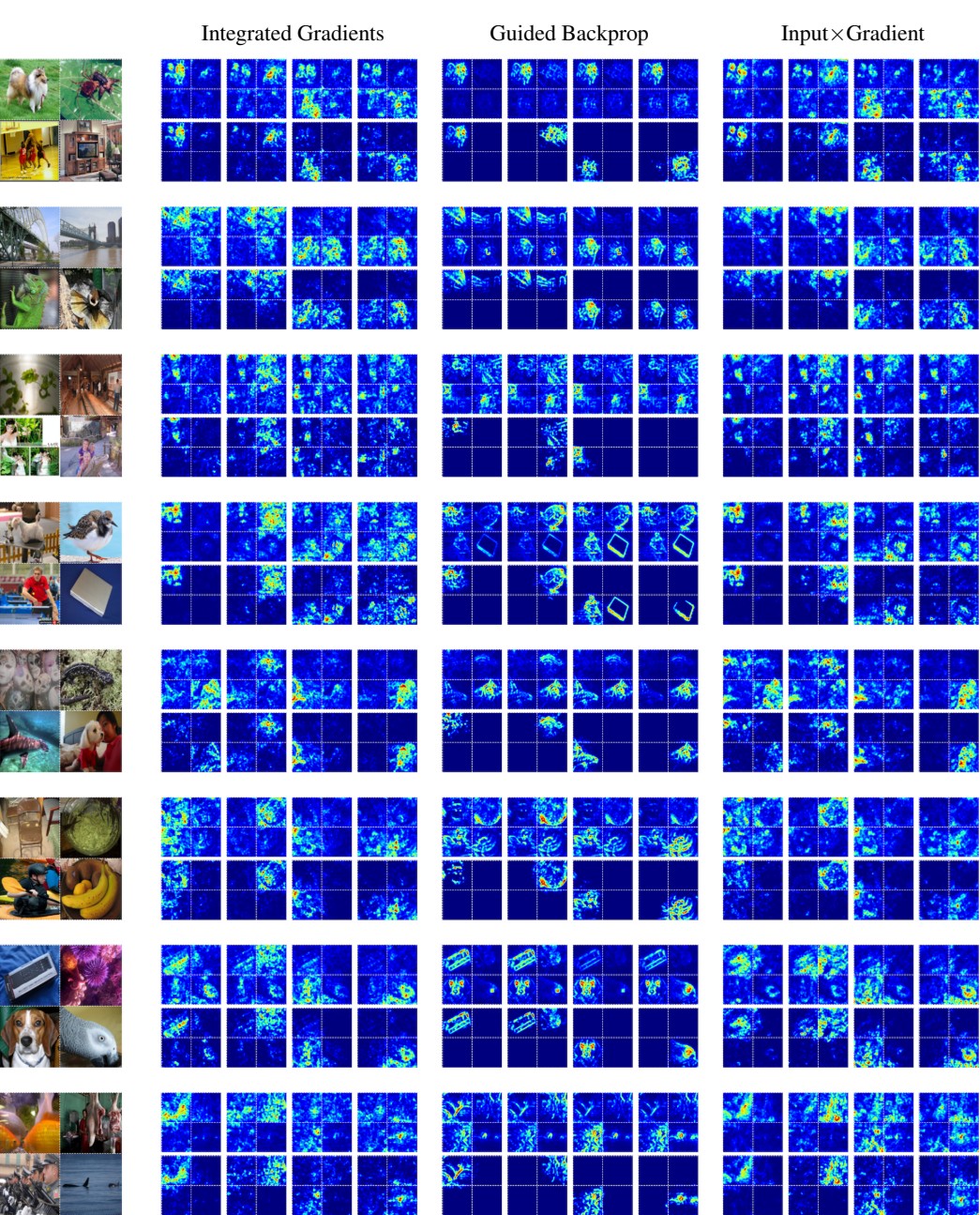

Figure 15: **ResNet50**: *VAR on the Grid Pointing Game.* We show examples from the grid pointing game for methods most affected by our framework (as columns: Integrated Gradient, Guided Backpropagation, Input×Gradient). Input Images are given on the left, for each we provide vanilla attribution methods (top row) and augmented with VAR (bottom row). For each, we show the attribution for the four different classes in the grid as columns.

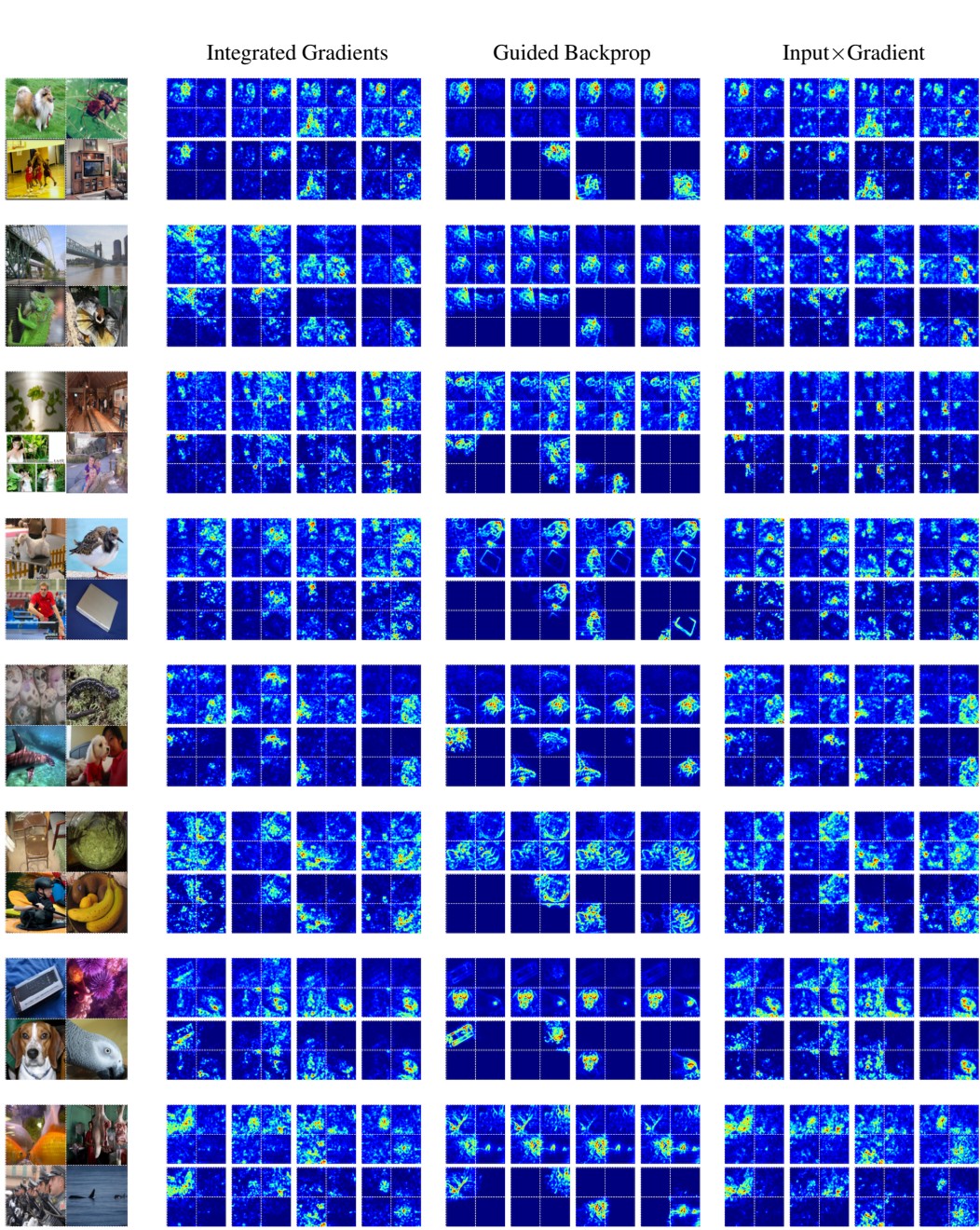

Figure 16: **DenseNet121**: *VAR on the Grid Pointing Game.* We show examples from the grid pointing game for methods most affected by our framework (as columns: Integrated Gradient, Guided Backpropagation, Input×Gradient). Input Images are given on the left, for each we provide vanilla attribution methods (top row) and augmented with VAR (bottom row). For each, we show the attribution for the four different classes in the grid as columns.

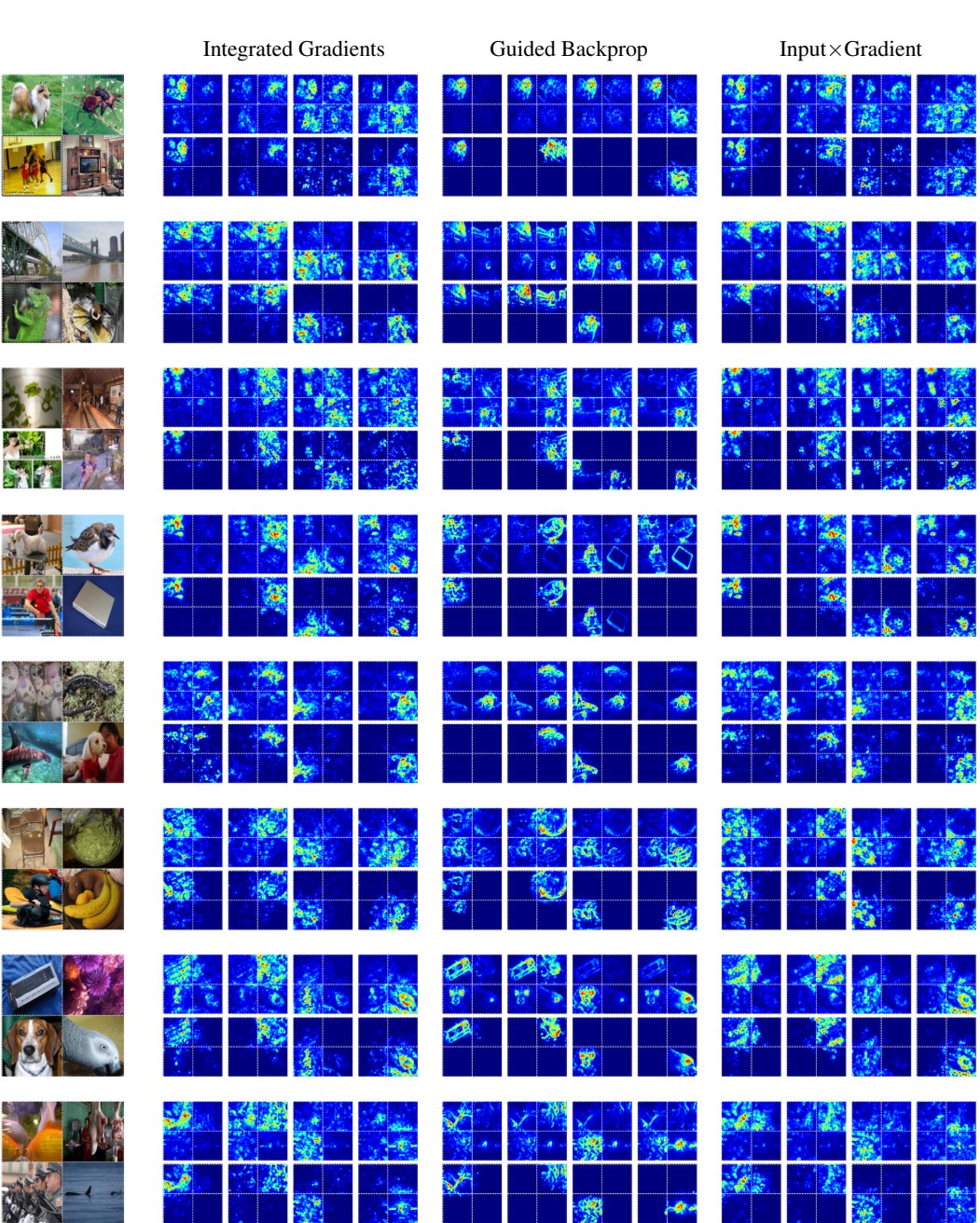

Figure 17: **WideResNet50-2**: *VAR on the Grid Pointing Game.* We show examples from the grid pointing game for methods most affected by our framework (as columns: Integrated Gradient, Guided Backpropagation, Input×Gradient). Input Images are given on the left, for each we provide vanilla attribution methods (top row) and augmented with VAR (bottom row). For each, we show the attribution for the four different classes in the grid as columns.

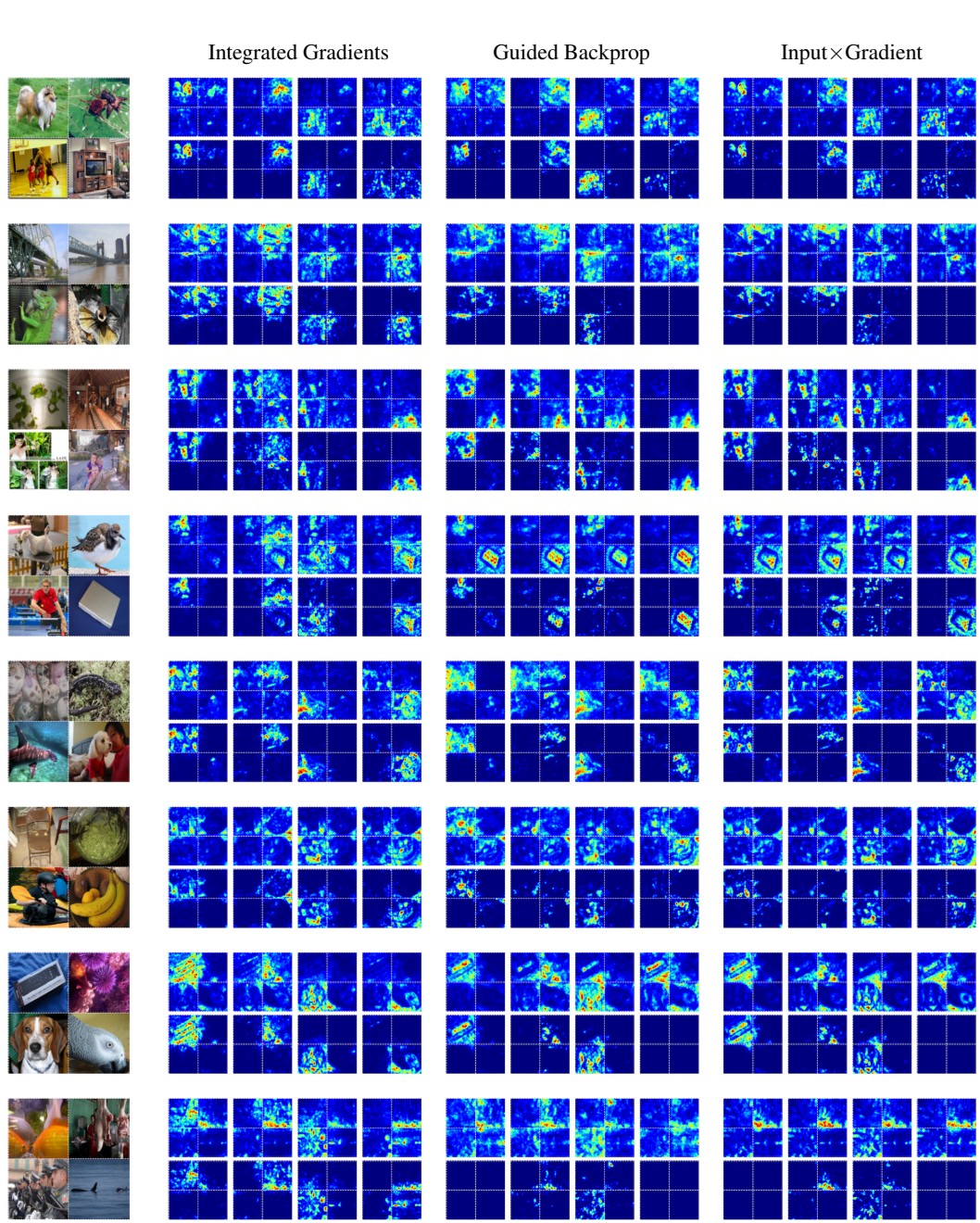

Figure 18: **ConvNeXt**: *VAR on the Grid Pointing Game.* We show examples from the grid pointing game for methods most affected by our framework (as columns: Integrated Gradient, Guided Backpropagation, Input×Gradient). Input Images are given on the left, for each we provide vanilla attribution methods (top row) and augmented with VAR (bottom row). For each, we show the attribution for the four different classes in the grid as columns.

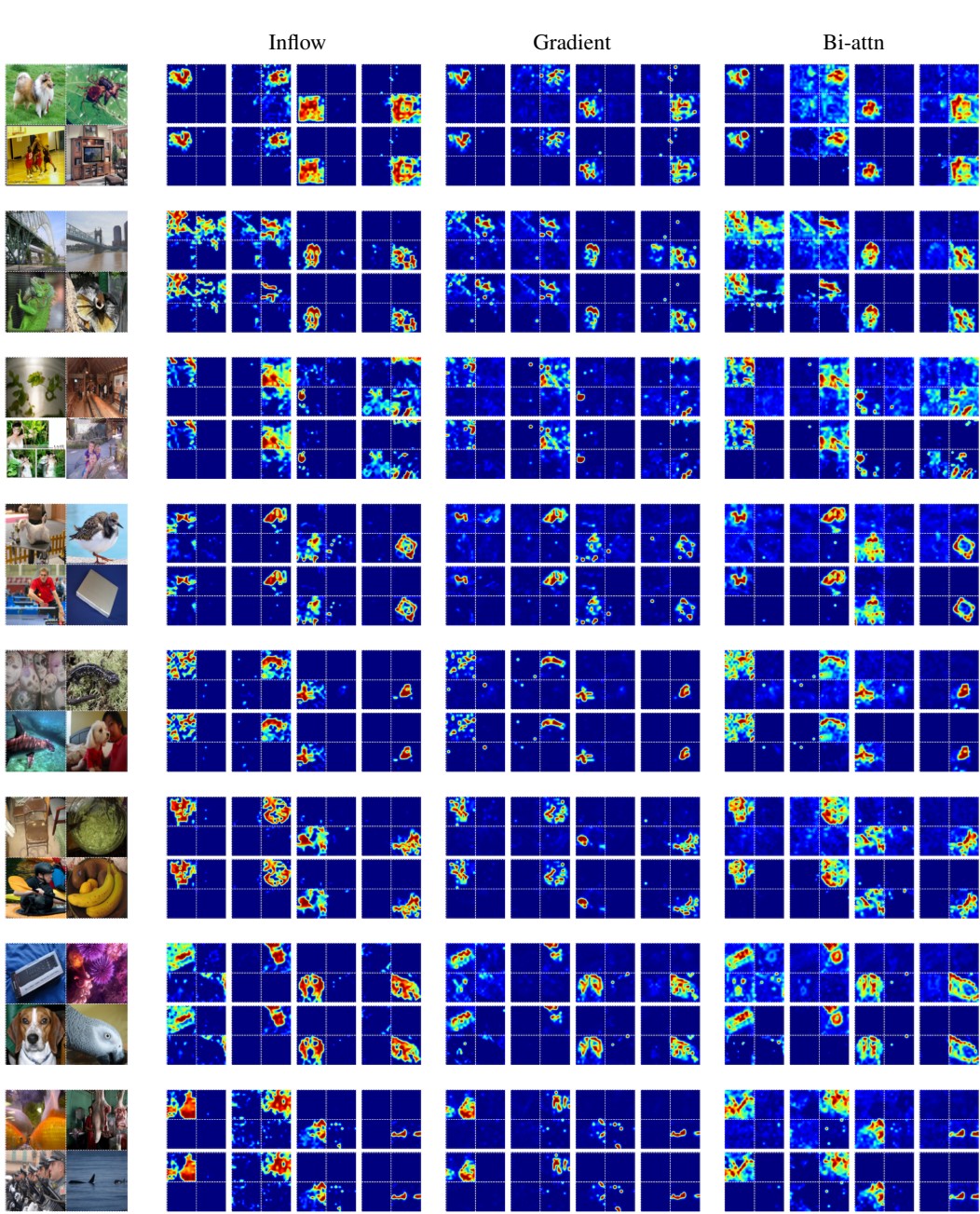

Figure 19: **ViT-base-8**: *VAR on the Grid Pointing Game.* We show examples from the grid pointing game for methods most affected by our framework (as columns: Integrated Gradient, Guided Backpropagation, Input×Gradient). Input Images are given on the left, for each we provide vanilla attribution methods (top row) and augmented with VAR (bottom row). For each, we show the attribution for the four different classes in the grid as columns.

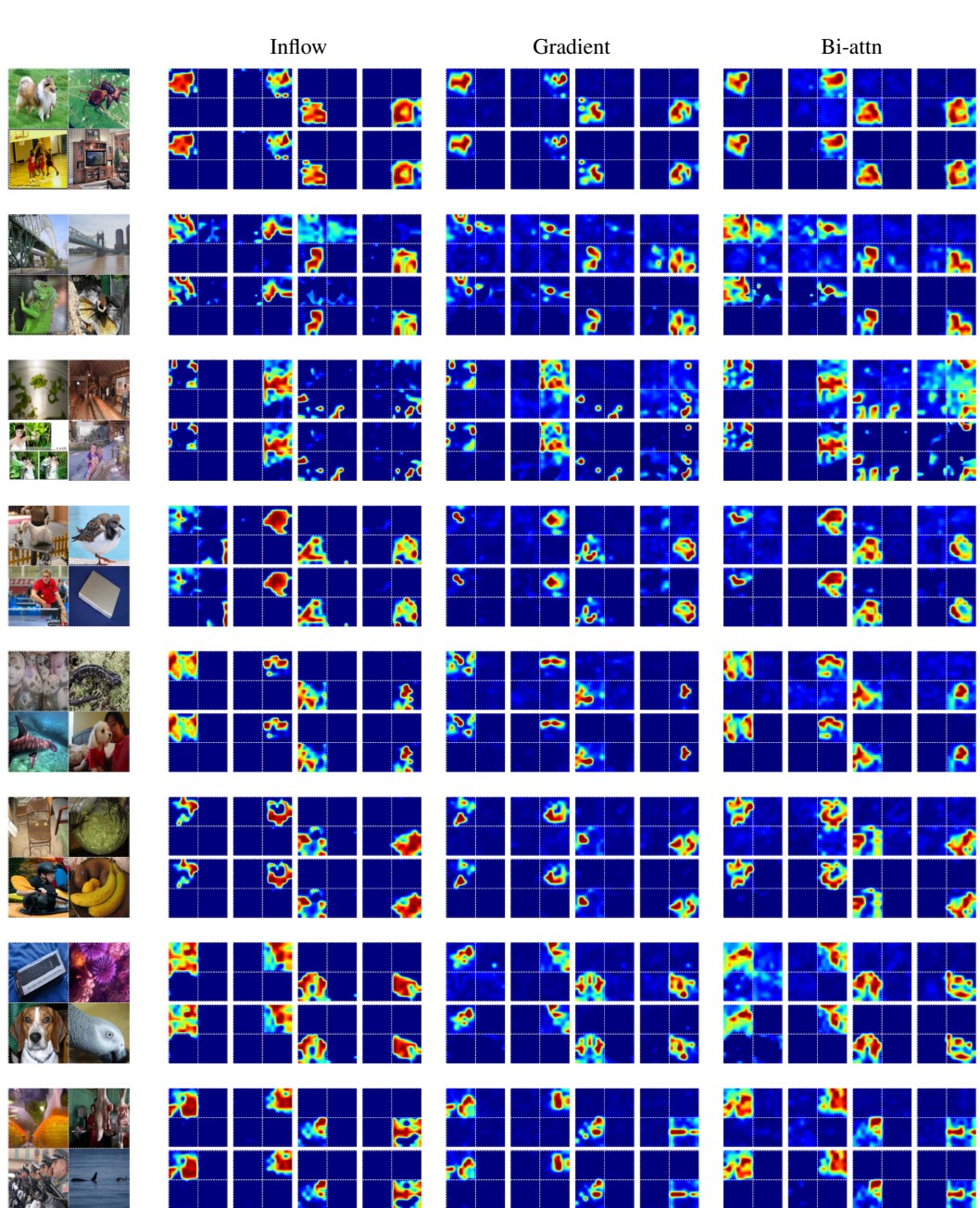

Figure 20: **ViT-base-16**: *VAR on the Grid Pointing Game.* We show examples from the grid pointing game for methods most affected by our framework (as columns: Integrated Gradient, Guided Backpropagation, Input×Gradient). Input Images are given on the left, for each we provide vanilla attribution methods (top row) and augmented with VAR (bottom row). For each, we show the attribution for the four different classes in the grid as columns.

Inflow          Gradient          Bi-attn

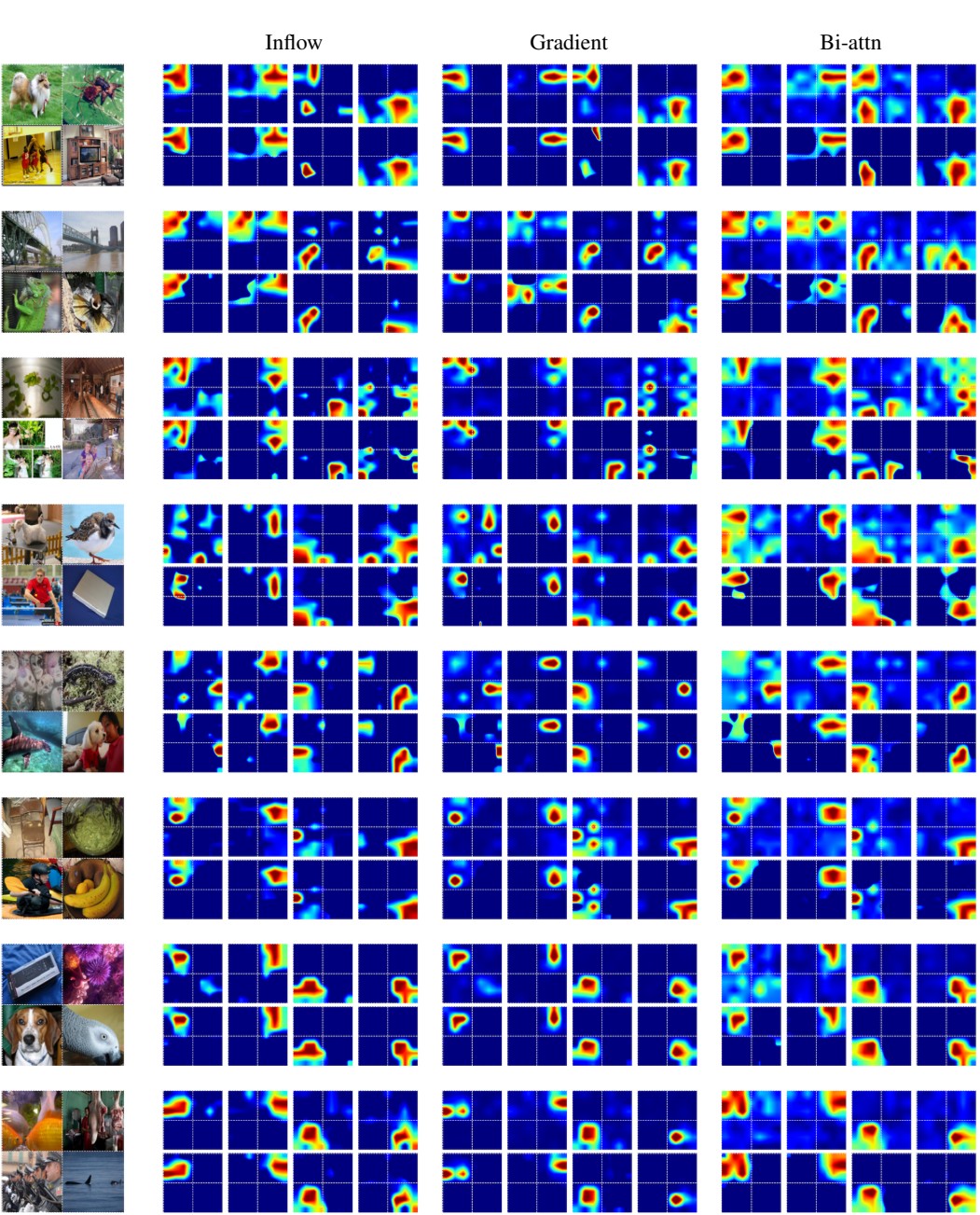

Figure 21: **ViT-base-32**: *VAR on the Grid Pointing Game.* We show examples from the grid pointing game for methods most affected by our framework (as columns: Integrated Gradient, Guided Backpropagation, Input×Gradient). Input Images are given on the left, for each we provide vanilla attribution methods (top row) and augmented with VAR (bottom row). For each, we show the attribution for the four different classes in the grid as columns.

