# OpenReview forum: "Now you see me! Attribution Distributions Reveal What is Truly Important for a Prediction"
_ICLR.cc/2026/Conference — ICLR 2026 Conference Withdrawn Submission_

### Official Review · Reviewer_NuzA · 2025-10-27

**Soundness:** 3
**Presentation:** 4
**Contribution:** 3
**Rating:** 8
**Confidence:** 3

**Summary:**

The authors introduce a method (VAR) that modifies existing attribution methods. The purpose of VAR is to provide increased clarity to the attribution heatmaps, further reducing noise and improving attribution localization. VAR uses a contrastive approach which compares the target logit against a set of other logits. The contribution of each pixel to the target class is modified by how strongly the set of contrastive classes also use each pixel in their attributions. The authors provide a wealth of qualitative results and strong quantitative results to back their proposed method.

**Strengths:**

- Originality: The authors are not introducing a new attribution method, but a way to improve existing methods. The method also takes into account issues with propagating through the softmax function. 3/4 on novelty.
- Quality / Clarity: Clear text and strong figures. Research quality is also high.
- Significance: Strong empirical results across all a number of different metrics. Good contribution.

Other Notes:
- Clever idea to to use other classes to inform the attribution of the target class
- Strong empirical results
- Low probability class suppression was a good idea

**Weaknesses:**

- Insertion test is included, but no deletion tests. Typically these tests are included together, so it is odd to include one and not the other.
- No run time tests included. While most attributions are computed offline, there are uses for attributions in real-time systems, so it would be good to have some results for those who wish to try to apply this in real time.
- No ablation results over the values / number of values used for temperatures. At present, it seems as though the numbers were arbitrarily chosen.
- Should add parameters used for each method evaluated to the appendix

**Questions:**

- Can the authors re-explain what they mean in lines 192-194 when describing equation 2? I think I understand, but I want to be sure.
- What is the runtime overhead for using this method? I assume it would be proportional to the number of classes used because you have to rerun the attribution method for each class.
- How did you choose the temperature values used by VAR? What are the effects of the size of the temperature values? What are the effects of the number of temperature values used?
- I see some discussion on how the set of classes is selected for use in VAR. However, no results are given based on the different selection approaches. I'd be very interested in seeing results around highly related classes. For example, Imagenet has a few different elephant classes. The VAR method does seem to nicely reduce noise and improve localization, so I'd be interested in seeing what features VAR pinpoints for tusker vs. african elephant vs. asian elephant. Also, how does the choice of classes impact quantitative results. Again, I know there is some discussion around this in the appendix, but including concrete results would definitely improve the significance of the paper.

Final Review:
The paper provides a strong method for significantly improving existing attributions. While the main negative is that it seems to impose a heavy runtime overhead, the method gives significant performance gains, grapples with a known problem for attributions, and the paper itself is high quality. I believe it should be accepted (8/10).

---

### Official Review · Reviewer_24H2 · 2025-10-28

**Soundness:** 2
**Presentation:** 3
**Contribution:** 3
**Rating:** 2
**Confidence:** 4

**Summary:**

This paper presents a contrastive method for better attributing inputs to model decision making. From the issue of current attribution approaches, where attributing from the logit to the input does not consider multiple classes, and the fact that you cannot effectively attribute from the softmax output to the input, this paper aims to create a new way to integrate information into an explanation to improve its class-discrimination. Specifically, their approach acts as a post-processing method which takes attributions not only at the target class, but multiple classes of the model output, and creates a new attribution from this set which extracts only target class-relevant features.

**Strengths:**

The paper is interesting and more-or-less easy to follow. The method is novel while being simple and having a clear mechanism that leads to its success.

There is a need for this method in the XAI field as it solves a well-known issue effectively with minor computational overhead.

The work has the potential to reveal interesting properties of attributions given proper analysis of how it does or does not improve their performance.

**Weaknesses:**

Formatting: Use \citep{} not \cite{} unless the citation is supposed to be read as part of the sentence. Lines 263-265 have repeated text due to an unremoved edited sentence. Figure 19-21 caption for attributions used is incorrect (references the CNN attributions not the ViT attributions). Multiple double citations (Hila Chefer, Jiamin Chen, Sukrut Rao, Chase Walker, Tingyi Yuan).

I find some discussion of the results to be missing. Mainly, I would like to have seen more information on why this approach fails to make improvements for some attributions. What does this behavior tell us about an attribution method or a model?

The experiments are well rounded in breadth of metrics and base methods that are improved, but I feel this paper should not be without a comparison to other methods altogether as it is now. This method is a pseudo post-processing approach in the way that it takes existing attribution methods and applies them across the classes of the model to make a new representation. While conceptually different, two post-processing approaches that come to mind are smoothgrad and XRAI. I think it is important to have these as basic baselines to understand how much more improvement your method yields (if this is the case).

To have a more well-rounded experimental section, it would have been valuable to see how your framework impacts a perturbation-based method (like feature ablation) which is builds explanations from the changes in the softmax output and may not require such interventions.

It is not clear what k is set to and what the best choice is. Appendix A outlines templates for defining the set, but I could not find a clear statement of what set size was used. It is obvious that k=4 for your predefined class sets on the 4-image pointing game, but what about the insertion and randomization tests? Does the best choice vary per dataset/model/method? Is it heuristic? Do you have ablation results to share? Since this is missing, it is hard to tell what the computational overhead is in a generalized application of the approach.

It is also confusing, and seemingly unnecessary, to mix $H$, $q$, and $\mathcal{H}$ the way that it is being done here.  $\mathcal{H}$ appears and seemingly redefines $q$ which was already a redefinition of $H$. These choices put undue burden on the reader. The substitutions and redefinitons, as used, do not aid reading comprehension.

**Questions:**

What is the intuition for the cases where this method does not create quantitative improvements in attribution methods such as Attn Grad (pointing game) and LRP (ins metric), the ConvNeXT model (ins metric), or InFlow in figures 19-21? (weakness 2)

Does it make sense to have no comparisons against other styles of attribution improvement? (see weakness 3).

How is the value of k chosen in insertion/randomization tests?

I urge the authors to use the lengthy rebuttal period to make changes to their document to answer these questions. My score will certainly improve to be favorable if these big issues can be fixed, as I think the methodology is strong and well worth publishing, but I cannot recommend the paper in its current form.

---

### Official Review · Reviewer_Vjjf · 2025-10-29

**Soundness:** 3
**Presentation:** 2
**Contribution:** 2
**Rating:** 4
**Confidence:** 5

**Summary:**

This paper proposes a multi-class attribution distribution mechanism aimed at enhancing the specificity and discriminative power of attribution methods in Explainable Artificial Intelligence (XAI). The core innovation lies in the introduction of VAR (Visualizing Actually Relevant Features), a technique that generates attribution results by computing the distribution of logit values across multiple classes, rather than analyzing only the logit of the predicted class. The method is architecture-agnostic and training-free, allowing flexible application to both CNNs and transformer-based architectures. The paper provides a comprehensive evaluation across various benchmarks, showing consistent improvements in localization accuracy, robustness, and overall attribution quality. However, the paper’s novelty primarily lies in engineering-level optimization rather than in theoretical advancement. Moreover, some interpretations of the experimental results are confusing, and the presentation and writing structure could benefit from further refinement.

**Strengths:**

1. The proposed VAR overcomes the limitation of traditional attribution methods that focus solely on a single-class logit. By introducing the softmax-based class competition concept into the attribution process, it constructs a cross-class attribution probability distribution at each pixel, thereby revealing both discriminative and shared features that the model truly uses for classification.
2. The proposed VAR can be seamlessly integrated with any existing attribution method, requiring no additional training and no dependency on specific network architectures, making VAR a simple yet effective plug-and-play improvement.

**Weaknesses:**

1. Although the proposed VAR method improves upon existing attribution techniques, its innovation primarily lies in engineering-level optimization, with limited theoretical depth and methodological novelty. It is suggested that the experimental section be further strengthened — particularly by expanding the diversity and evaluation of attribution methods. The authors could include additional comparisons with widely used approaches such as Shapley Value and DeepLIFT, to better validate the effectiveness of the proposed method across broader application scenarios and enhance the paper’s depth and persuasiveness.
2. There is some confusion in the experimental results. In Section 4.3 *“Sanity Checks”* (Figure 6), the y-axis represents the similarity between attribution maps before and after randomization, while the x-axis represents the proportion of layers randomized. When the randomization ratio is 0%, the model remains unperturbed, so the similarity should be 100%. However, the results shown in the figure do not seem to reflect this. In addition, some curves in Figure 6(b) (e.g., *Inflow + VAR*) are already close to zero at 0%, which requires further explanation. The authors are encouraged to provide a more detailed analysis and clarification in the revision.
3. The paper has several typesetting and writing issues. There are multiple spelling and formatting errors (e.g., the misspelling of *“ADDITIONAL RESULST”*), inconsistencies in citation formatting (such as inconsistent capitalization of conference names), and some equations without numbering. In addition, several figures (e.g., *Figure 2*’s structural diagram) appear rough and not sufficiently clear. The authors are advised to refine these elements to improve readability and presentation quality.
4. Some figures are not explicitly referenced in the main text. For example, *Figure 3* and *Figure 5* are not mentioned or discussed in the corresponding sections, making it unclear how they relate to the main arguments. The authors should ensure that all figures are properly cited and explained in the text to maintain consistency and logical coherence between the figures and the narrative.

**Questions:**

Please address the questions raised in the Weaknesses section.

---

### Official Review · Reviewer_odkX · 2025-11-01

**Soundness:** 2
**Presentation:** 2
**Contribution:** 3
**Rating:** 4
**Confidence:** 3

**Summary:**

Common attribution methods utilize the raw logits of neural networks and deliberately ignore the final softmax layer to avoid vanishing gradients. While this circumvents numerical instabilities, it also reduces faithfulness: by looking at the unnormalized logits, attributions may overemphasize regions that are dominated by competing logits. This paper addresses this limitation by retaining conventional attribution methods, but adding a post-processing step that normalizes the resulting attribution maps across (a subset of) classes on a per-pixel basis, similar to a conventional softmax. This modification (“VAR”) naturally suppresses pixels dominated by other classes, resulting in more class-discriminative attribution maps.

For evaluation, the paper presents localization, insertion, and randomization-based analyses, following standard protocols for attribution map assessment. The baselines are the respective attribution methods without the proposed VAR modification. For localization, quantitative and qualitative experiments show large improvements over the baselines on all three evaluation variants, with a visually and numerically apparent improvement in object focus. The insertion and randomization evaluations show consistent, but more moderate gains compared to the non-VAR baseline.

**Strengths:**

The proposed modification is a conceptually very simple, but interesting and seemingly novel. It is shown to be effective in making the localization of the attribution maps more class-discriminative. The evaluation is fairly standard and support most claims of this paper (though see weaknesses below), and cover both CNN- and ViT-based architectures. The set of examined attribution methods is comprehensive, and the significant gain across the board strengthens the confidence in the provided attribution map and pointing game results. While some of the figures are hard to read, the presented qualitative results are mostly convincing and support the claim of the paper.

Notably, the proposed modification is agnostic regarding the attribution method and thus applicable to future approaches, as long as they assign unnormalized attribution values to individual pixels. Ideally, this allows generalizing beyond the scope of the conducted evaluation.

**Weaknesses:**

The presentation needs significant work, especially regarding clarity. The paper appears to have been put together hastily, with duplicate sentence fragments (l. 263ff), broken citations (always using \citet even when parenthetical citations should be used), and missing information. The provided figures are difficult to read: the teaser figure is not informative and, e.g., does not clearly indicate which maps correspond to the proposed method (the reference in l. 068ff to Fig. 1 can only be understood after having read the rest of the paper). Figures 3 and 4 lack the baseline label and a clear structure to guide the reader, which makes it hard to quickly understand what is being compared.

One significant nuance that is not discussed is how the proposed contrastive attribution deals with contextual information and whether it even allows to reveal the use of context in the classifier. The paper tacitly assumes that attribution maps should only highlight pixels that are “on the object”, but it is not actually clear whether this is faithful to the model as the model may be using context to arrive at a certain classification decision. The contrastive modification in VAR suppresses pixels where other classes dominate, but this is almost always the case for context. This means that the proposed VAR approach actively suppresses the attribution of contextual information. Whether this is desirable is not all that obvious.

A related issue lies with the evaluation. It is fairly obvious that a clearer object-focus, as is the result from applying the VAR approach, will be rewarded in the grid pointing game. But since the non-target examples do not provide relevant context for the target one, we cannot assess whether the proposed method will be faithful in regards to the use of context in the image classifier. To rectify this, it would be preferable to use a synthetic benchmark dataset like FunnyBirds (Hesse et al., ICCV 2023) that supplies ground-truth object (part) importances to provide a more complete picture.

The role of two hyperparameters remain underexplained: the paper notes that averaging over temperatures t ∈ {1, 5, 100} achieves visually more pleasing results, but it is unclear how much this particular choice impacts both the qualitative and quantitative results. The same applies to the attribution suppression threshold (≈1/K). How dependent are the results on this particular value?

The impact on the computational effort has not be quantified. Having to compute attributions for a whole (sub)set of classes surely increases the computational effort, but it’s not clear by how much.

Related, it has not become particularly clear how the contrastive set of classes $\mathcal{C}’$ is chosen for each dataset. Appendix A.1 contains some general discussions regarding this, but unless I overlooked something, it has not been specified exactly how the classes are chosen in practice (l. 319 is very vague in this regard). The actual choice of contrastive classes is not only relevant for understanding but also for assessing the computational overhead (see above).

Finally, while the introduced modification is intentionally simple - which is acceptable given the provided results - the presentation often feels more complicated and confusing than necessary (e.g., using three different notations for a standard softmax in 3.2). Streamlining the notation would improve readability. Likewise, for being the primary visualization of the approach, Figure 2 does little to support the reader in understanding the method.

Minor points:
* l. 168: This is not really “Computer Vision”… The paper considers image classification, which is a highly simplified proxy that is not particularly representative of real computer vision tasks. I suggest rephrasing.
* l. 191: Why only use gradients to define the attribution? What about other attribution methods?

**Questions:**

* Could you clarify the use of the proposed approach in conjunction with the use of context in the image classifier? Would the proposed approach be faithful in this case?
* How has the contrastive set of classes been choses for each dataset? How many classes are being considered and what is the computational overhead?
* Would the proposed approach also work well in evaluations where ground truth enables an assessment of the faithfulness (e.g., FunnyBirds)?
* Please include a small ablation over temperature choices: how would localization change (qualtitatively and quantitatively) when not using temperature smoothing, and when varying $t$?
* Likewise, provide an ablation on the attribution pixel-suppression threshold (currently ≈1/K): how sensitive are localization, insertion, and randomization metrics to this setting?

---

### Note · Authors · 2025-11-13

**Comment:**

Dear reviewers,

We thank you for your thoughtful feedback and for recognizing both the strong empirical gains of VAR and its broad applicability across architectures and attribution methods. We agree with the points raised about class-set selection, hyperparameters, and runtime, which are valid and could be addressed.

We do not expect to be able to incorporate all the suggested changes to pass this review cycle. We are therefore withdrawing the submission and are very grateful to the reviewers for their constructive input, which we will use to update the work.

Best regards,
the authors of VAR

**Withdrawal Confirmation:**

I have read and agree with the venue's withdrawal policy on behalf of myself and my co-authors.